# RoboMemory: A Brain-inspired Multi-memory Agentic Framework for Interactive Environmental Learning in Physical Embodied Systems

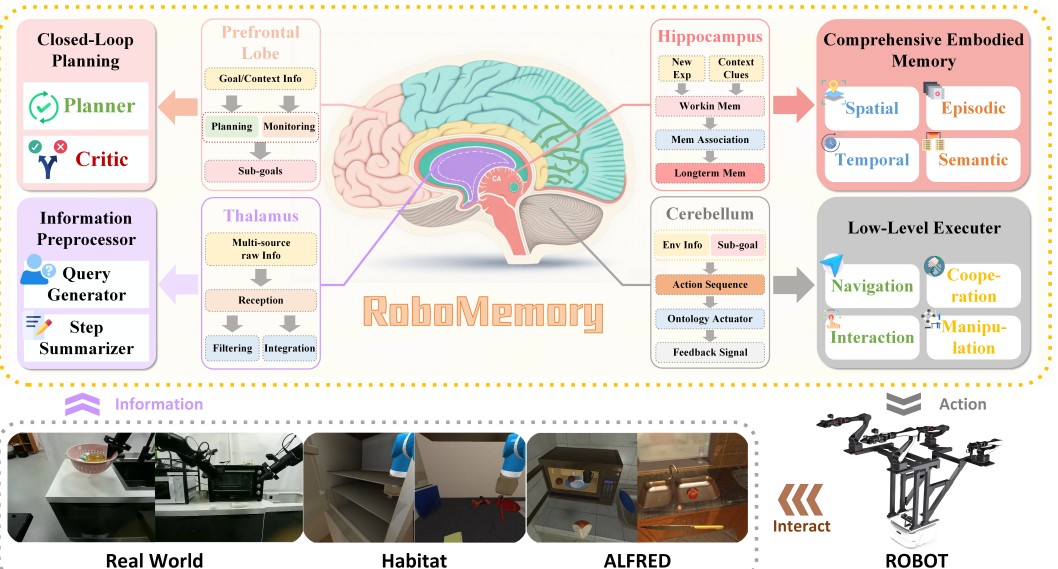

Figure 1: RoboMemory adopts a brain-inspired architecture that maps neural components to agent modules, enabling long-term planning and interactive learning across diverse environments (real-world, Habitat, ALFRED) and robotic hardware.

## ABSTRACT

Embodied agents face persistent challenges in real-world environments, including partial observability, limited spatial reasoning, and high-latency multi-memory integration. We present RoboMemory, a brain-inspired framework that unifies Spatial, Temporal, Episodic, and Semantic memory under a parallelized architecture for efficient long-horizon planning and interactive environmental learning. A dynamic spatial knowledge graph (KG) ensures scalable and consistent memory updates, while a closed-loop planner with a critic module supports adaptive decision-making in dynamic settings. Experiments on EmbodiedBench show that RoboMemory, built on Qwen2.5-VL-72B-Ins, improves average success rates by 26.5% over its baseline and exceeds the closed-source state-of-the-art (SOTA) Claude3.5-Sonnet by 1%. Real-world trials further confirm its capacity for cumulative learning, with performance improving across repeated tasks. These results highlight RoboMemory as a scalable foundation for memory-augmented embodied intelligence, bridging the gap between cognitive neuroscience and robotic autonomy.

## 1 INTRODUCTION

Recent advances in Vision-Language Models (VLMs) (Hurst et al., 2024; Bai et al., 2025) have enabled their growing use in embodied tasks (Park et al., 2023; Hu et al., 2023). VLM-based embodied agents can process multimodal inputs and generate high-level textual commands (e.g., "Pick

up the cup"), which require translation via tool APIs to become executable robot actions. In contrast, Vision-Language-Action models (VLAs) (Kim et al., 2024; Black et al., 2024; Bjorck et al., 2025; Chi et al., 2023) produce low-level control signals directly but generally rely only on the latest observation. This limits their ability to perform long-horizon, multi-step tasks that require reasoning over task history. In summary, VLA models enable direct robot control but lack high-level planning capabilities, and VLM-based embodied agents support strategic planning but struggle with direct motor control. This highlights a key gap inherent in two distinct technical approaches to embodied intelligence.

To bridge this gap, recent work (Yuan et al., 2025; Shi et al., 2025; Tan et al., 2025) proposes a "VLM planner + VLA executor" paradigm. Here, VLM-based embodied agents serve as high-level planners that decompose complex tasks (e.g., "make a coffee") into executable sub-instructions (e.g., "grasp the cup") that VLAs can complete. Although this paradigm improves performance on multi-step tasks, prior work suffers from two key limitations in real-world settings. First, real-world tasks (e.g., kitchen operations) require navigating across multiple locations to gather objects and tools, but the environment remains only partially observable at any time due to robots' limited field of view and dynamic occlusions. This necessitates a planner with robust spatial awareness and long-term memory to maintain a consistent spatial awareness across viewpoints. However, most VLM-based agents rely on chat-style context windows (e.g., logging instruction – feedback pairs (Yao et al., 2022)), which lack mechanisms for maintaining an overview of the environment's spatial layout. Consequently, agents cannot reliably track object locations or recognize previously visited states. Second, pretrained VLMs are rarely trained on embodied planning trajectories, especially long-horizon, spatially grounded ones. So VLM-based agents often struggle to generalize to real-world settings (Yang et al., 2025a). To overcome these challenges, VLM-based planners must support interactive environmental learning — the ability to acquire, integrate, and retrieve spatial, episodic, and semantic knowledge during task execution, thereby enabling adaptation through experience.

To support long-horizon planning and interactive learning in real-world settings, agents require a comprehensive memory system with multiple specialized modules. Recent frameworks (Tan et al., 2024; Glocker et al., 2025; Wang et al., 2023; Agashe et al., 2024; Fu et al., 2024a; Zhao et al., 2024; Chen et al., 2024a) have integrated Retrieval-Augmented Generation (RAG)-based memory to enhance planning and interactive environmental learning (Gao et al., 2023), but most are designed for simulated environments. A key limitation for real-world deployment is the absence of spatial memory, which is critical for building spatial awareness and providing context for planning. Additionally, existing multi-module memory systems often incur significant inference latency.

To overcome these limitations, especially the need for memory that is efficient, spatially grounded, and persistent in dynamic environments, we return to the essence of intelligence — how does the human brain plan, remember, and learn in dynamic environments? Inspired by cognitive neuroscience, we have designed RoboMemory, a parallel multi-memory architecture that simulates key functional regions of the brain. RoboMemory features a hierarchical and parallelized architecture enabling long-term planning and continuous adaptation. Drawing inspiration from cognitive neuroscience (Milner, 1998), RoboMemory comprises four core components (Figure 1): (1) Information Preprocessor (thalamus-inspired) for multimodal sensory integration. (2) Comprehensive Embodied Memory System (hippocampus-inspired), which organizes experiential and spatial knowledge through a three-tier structure (long-term, short-term, and sensory memory). Within this tiered system, four memory modules: Spatial, Temporal, Episodic, and Semantic operate under a unified, parallel-update paradigm to enable coherent knowledge integration while minimizing latency. (3) Closed-Loop Planning Module (prefrontal cortex-inspired) for high-level action sequencing. These three modules provide a high-level planner with comprehensive sensory and memorization ability. (4) Low-level Executor (cerebellum-inspired), consisting of a VLA-based operation model and a SLAM-based navigation model. The Low-level Executor directly controls the robot with low-level control signals to navigate and operate in the real-world environment.

To verify whether RoboMemory truly addresses the problems of long-horizon planning and interactive learning, we evaluate RoboMemory on EmbodiedBench, a long-horizon planning benchmark (Yang et al., 2025a). Using Qwen2.5-VL-72B as the base model, RoboMemory improves average success rates by 26.5% over its base model and 1% over the closed-source state-of-the-art model, Claude3.5-Sonnet Anthropic (2024). In real-world trials, RoboMemory executed diverse tasks twice consecutively: once for environmental familiarization (learning phase) and once for memory-augmented execution (testing phase) without resetting memory. The observed performance

improvement validates RoboMemory's capacity for interactive environmental learning. We further conduct ablation studies and error analysis to quantify component contributions and identify remaining limitations. We summarize our contribution as follows:

- We propose a brain-inspired unified embodied memory system, integrating four concurrently updated modules (Spatial, Temporal, Episodic, Semantic) into a single framework. It enables efficient, comprehensive memory operations and coherent knowledge integration, which are critical for interactive environmental learning in real-world embodied scenarios.

- We design a retrieval-based incremental update algorithm for real-time evolution of Spatial Knowledge Graphs (KGs). By retrieving relevant subgraphs, detecting local inconsistencies, and merging new observations, it ensures efficient, consistent KG maintenance and addresses the scalability bottleneck of previous KG-based methods in embodied settings.

- RoboMemory supports interactive environmental learning for real-world physical robots: it enables sequential diverse tasks without memory reset, with experience accumulation driving steady performance improvements, demonstrating practical long-term autonomous learning in physical scenarios.

## 2 RELATED WORK

### 2.1 VLM/LLM-BASED AGENTIC FRAMEWORKS IN EMBODIED TASKS

The rapid advancement of VLMs/LLMs has led to diverse agent frameworks in embodied environments (Yao et al., 2022; Song et al., 2023; Lin et al., 2024). Embodied tasks involve partial observability and long-horizon planning, requiring memory systems to retain context. Some use time-ordered context buffers for short-term memory (due to VLMs/LLMs' limited long-context processing) (Yao et al., 2022; Packer et al., 2023); others adopt experience buffers as long-term semantic memory (Fu et al., 2024a; Shinn et al., 2024). For long-duration tasks, skill libraries serve as procedural memory, with agents accumulating skills via interaction (Wang et al., 2023; Tan et al., 2024). However, in real-world settings, the low-level executor may fail to complete the task, making it challenging to construct a reusable, code-based skill library. So, explicit procedural memory still needs to be improved in real-world settings. Moreover, Recent efforts integrate diverse memories (Zhang et al., 2023; Tan et al., 2024; Agashe et al., 2024) but focus on virtual/GUI environments, leaving real-world multi-modal memory support for long-term planning under-explored.

### 2.2 VISION LANGUAGE ACTION MODEL

Current work on VLA models uses imitation learning to output low-level controls from language and visuals (Black et al., 2024; Zhao et al., 2023; Bjorck et al., 2025; Kim et al., 2024) but is limited to tabletop tasks and single actions, restricting long-horizon planning. VLAs lack long-term execution abilities, while high-level agents excel at planning. Recent works combine high-level frameworks with VLA executors, some augmented with simple memory (Shi et al., 2025; Tan et al., 2025; Yuan et al., 2025; Yang et al., 2025b) for longer tasks. However, real-world robots need more sophisticated memory to handle continuous multi-task operations over extended periods.

### 2.3 MEMORY FRAMEWORKS

Many previous works improve long-term planning via memory systems: Voyager (Wang et al., 2023) uses a skill library in Minecraft but lacks diverse memory types; CoELA (Zhang et al., 2023) includes procedural, semantic, and episodic memory with a task-specific 2D map; MSI-Agent (Fu et al. 2024a) utilizes insight as long-term memory for in-task learning. Hippo Retrieval Augmented Generation (RAG) (Gutiérrez et al., 2024) mimics the hippocampus and introduces KGs as long-term memory indices (Burgess et al., 2002; Chen et al., 2020), enhancing retrieval. However, the previous approach is mainly focused on constructing a KG with a static long context, such as a book, but it is hard to update the graph. We need to update the information in KG for the embodied task. Our approach builds a more general LLM-based memory system using a dynamic KG like Hippo RAG, which is designed for embodied tasks. Furthermore, we summarize the differences among different memory systems in previous work. The comparison is shown in the Table 3 in Appendix D.

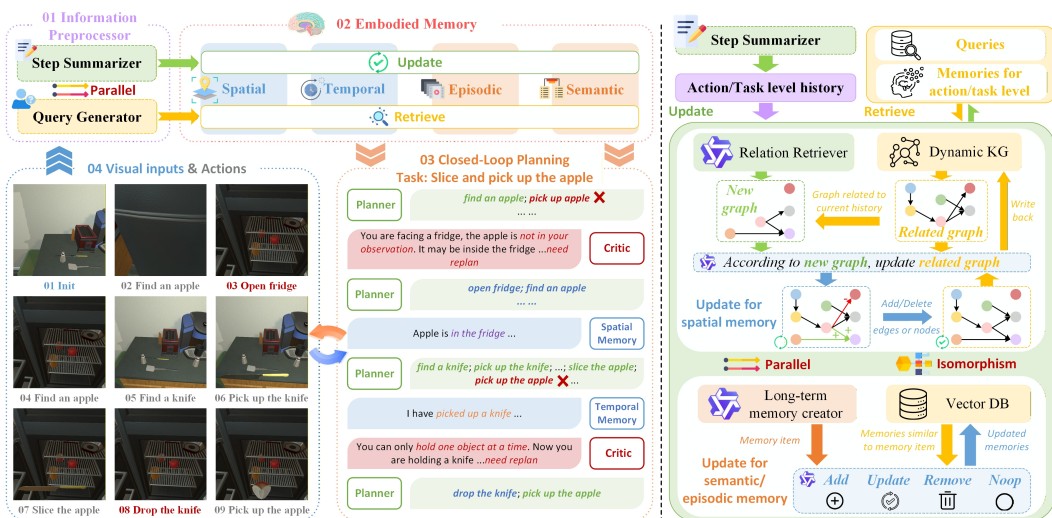

Figure 2: (a) Left: The loop where the Planner, Critic, and Embodied Memory interact to adjust plans based on real-time visual inputs. Colored text denotes the execution status of actions (success/rejected/replanned). (b) Right: Spatial memory maintains a relevance/similarity-updated KG, and Semantic/Episodic memory manages a Vector DB with analogous logic. Besides, Temporal memory is implemented as a linear FIFO buffer that stores step-wise summaries generated by the Step Summarizer.

## 3 ROBOMEMORY

RoboMemory is a hierarchical embodied agent system that equips robots with three core memory capabilities: historical interaction logs, dynamically updated spatial layouts, and accumulated task knowledge. As illustrated in Figure 2, each iteration, RoboMemory follows a process of "Perception – Memory – Retrieval – Planning – Execution" process, ensuring that the agent continuously calibrates its memory and behavior in dynamic environments.

First, the information preprocessor converts multimodal sensor inputs into a textual summary of the current scene, which serves as the primary input to the Comprehensive Embodied Memory. Next, the Comprehensive Embodied Memory updates its internal representations, including action histories, object locations, and experiential knowledge. After information is updated, the memory system retrieves contextually relevant entries to inform the Closed-Loop Planning Module. Then, leveraging this contextual memory, the Closed-Loop Planning Module generates high-level, text-based action instructions. Finally, these commands are dispatched to low-level executors, who will directly control the robot and complete the instructions. The algorithm is demonstrated in Appendix B.

### 3.1 INFORMATION PREPROCESSOR

At each time step $t$, RoboMemory receives a visual observation $\mathcal{O}_t$: an RGB frame (in simulation) or a short video clip (on physical robots), representing the agent's observations.

Since raw visual data is unsuitable for direct use in memory construct and retrieval, RoboMemory first employs an information preprocessor to convert multimodal observations into textual representations, thereby providing a semantic interface for subsequent memory and planning modules. The information preprocessor executes two Vision-Language Models (VLMs) in parallel: (1) *Step summarizer* $\mathcal{S}$: Transforms $\mathcal{O}_t$ into a concise textual description $s_t$ of the just-executed action. The string $s_t$ is stored in the system's working memory. (2) *Query generator* $\mathcal{Q}$: Derives a list of queries $q_t = [q_t^{(1)}, q_t^{(2)}, \ldots, q_t^{(N)}]$ from the same observation $\mathcal{O}_t$. Each query $q_t^{(i)}$ is a natural language-based query. These queries are used to query information from the memory system that may be useful.

Together, $\mathcal{S}$ and $\mathcal{Q}$ provide a swift, text-based interface between raw sensory data and provide basic information in each iteration for RoboMemory's Comprehensive Embodied Memory System.

### 3.2 COMPREHENSIVE EMBODIED MEMORY SYSTEM

To address the long-term memory limitations in current embodied agent frameworks, we propose the Comprehensive Embodied Memory System. This system consists of multiple memory modules. We denote the memory system containing $L$ distinct modules as $M_t = [M_t^{(1)}, M_t^{(2)}, ..., M_t^{(L)}]$, where $M_t$ represents the memory stored at step $t$, and $M_t^{(l)}$ denotes the $l$-th memory module. Generally, the memory update and retrieval process of iteration $t$ is shown below:

$$M_t = \mathcal{U}(M_{t-1}, s_t) \tag{1}$$

$$r_t = \mathcal{R}(M_t, q_t) \tag{2}$$

First, we update memory modules with algorithm $\mathcal{U}$, where we update the previous memory $M_{t-1}$ using the latest summarization $s_t$. Then, with updated $M_t$, we use an algorithm $\mathcal{R}$ to retrieve the information that is useful for the planning module. In $\mathcal{R}$, We use queries $q_t$ to query $M_t$, yielding retrieval results from each module: $r_t = [r_t^{(1)}, r_t^{(2)}, ..., r_t^{(L)}]$. These results are then passed to the planning module, which helps it plan future movements. However, sequentially updating and retrieving from $L$ modules would be a slow process. Therefore, we parallelize these steps across all modules, which significantly enhances the system's efficiency.

In implementation, the memory system consists of four distinct modules ($L = 4$): Temporal Memory, Spatial Memory, Semantic Memory, and Episodic Memory. For efficiency, all memory modules are updated and retrieved in parallel. Thus, even with multiple modules, the system remains highly efficient. Functionally, inspired by cognitive psychology Liu et al. (2025), our modules handle memory at different levels. In cognitive psychology, memory is divided into Sensory Memory, Short-term Memory, and Long-term Memory. Mirroring this hierarchy, our modules are organized as follows. First of all, the Information Preprocessor's $\mathcal{S}$ summarizes the agent's interactions with the environment at each iteration. It acts as Sensory Memory. Secondly, Temporal Memory and Spatial Memory function as short-term memory. These two memories will update at each iteration. They are designed to store the information of sensory memory in every iteration. For Temporal Memory, we record the agent's action history sequentially, while for Spatial Memory, we dynamically record the spatial relationships between different objects in the environment based on Sensory Memory in each iteration. These memories can provide a relatively long and detailed history of the current task for the Agent to make a future plan. Thirdly, Semantic Memory and Episodic Memory serve as Long-term Memory. They update only when meaningful information arises (e.g., after task completion). These memories store highly abstract knowledge, not limited to the current task, but synthesized from past experiences. This knowledge—factual, event-based, and experiential—improves the agent's future task performance. It is the source of RoboMemory's interactive learning capability. We now detail each module.

**Temporal Memory.** In the Temporal Memory, we record Interactions between the robot and the environment (i.e., Sensory Memory) of each iteration sequentially. This information can provide the embodied agent with simple awareness of "What I have done". For such temporally sequential memory, a simple structure is sufficient: a sequential buffer with automatic summarization triggered when the record sequence reaches its capacity. In a specific design, temporal memory can store up to $N$ interaction summaries, each generated by an information preprocessor. When the buffer is full, we compress the oldest $N$ steps into a single summarized entry using a VLM, which is then reinserted at the front of the buffer, ensuring continuous context retention without unbounded growth. However, the information from previous memories will gradually be lost as we summarize it multiple times. For retrieval, we provide all existed memory in text to downstream modules.

**Spatial Memory.** The spatial memory is designed to dynamically record the high-level spatial relationships of different entities in the environment. However, Current spatial memory approaches often rely on RGB-D cameras to reconstruct 3D point clouds (Zhang et al., 2023; Chang et al., 2023). These representations are too detailed for high-level planning in embodied agents. For example, Precise geometric relationships (e.g., exact distances between objects) are unnecessary.

To address these problems, inspired by Gutiérrez et al. (2024), we use Dynamic KG to store high-level spatial information: objects and positions in the environment become vertices of KG, and spatial relations between objects or positions are encoded as edges. The KG focuses on high-level spatial relations (e.g., "cup on table", "key left of drawer"). By these settings, spatial KG focuses on semantically meaningful, task-relevant relations. This spatial information enhances the agent's spatial reasoning capability in dynamic environments.

However, as related work shows, most KG construction algorithms are designed for static long content. This does not meet the demand of using KG as spatial memory for the agent. The KG needs to update efficiently in response to new information. To address this issue, we introduce a retrieval-driven, incremental KG update algorithm that maintains a locally modifiable, globally consistent, and dynamically adaptive spatial memory. As illustrated in the right panel of Figure 2, the update process proceeds in four steps: (1) retrieves the most relevant sub-KG around new observations. (2) Injects new relations from the current observation by a VLM-based Relation Retriever. (3) Detects and resolves conflicts between newly extracted relations and existing ones (e.g., "cup on table" vs. 'cup in drawer") using a VLM-based resolver, which decides whether to add, delete, or modify edges. (4) Merges back and prunes isolated vertices. Moreover, our retrieval-based incremental update algorithm is accompanied by provable efficiency guarantees. For a KG with $n$ vertices and maximum degree $D$, the number of vertices processed per update is bounded by $O(D^K)$, where $K$ is the retrieval hop distance (see Appendix E.3 for formal analysis). Further architectural and implementation details are provided in Appendix E.1.

**Semantic Memory.** In cognitive psychology, semantic memory stores time-independent facts. These facts are stable, update slowly, and require long-term retention. In RoboMemory, semantic memory records task-relevant experiences and environmental knowledge during execution. This information can help RoboMemory adapt to new environments or tasks. This information is highly abstract and does not need to be updated frequently. In RoboMemory, Semantic memory updates when new information is encountered during execution, for example, after completing a subtask or encountering important information. To store and update memories efficiently, we design a memory management system based on a vector database. In the vector database, each experience/fact is described in natural language (denoted as a memory item). Each memory item is converted into a semantic vector for querying. For dynamic updates, we adapt a framework from prior work Chhikara et al. (2025). As shown in the bottom-right of Figure 2, the semantic memory update algorithm involves two VLM-based modules. Firstly, Long-term Memory Creator generates new memory items based on short-term memory. We retrieve the top-$S$ most similar existing memory items from existing memory via cosine similarity. A VLM-based updater then compares new and existing items to decide whether to: *add* the new item, *update* an existing item, *remove* an outdated item, or perform *Noop* (if redundant). Since updates only involve a maximum of $S$ previous memory items and the update process is parallelized across all memory modules, this update method ensures that semantic memory remains efficient even as the database grows. For retrieval, we use the same process as a traditional vector database. We use queries to extract top-$N$ relevant information for downstream modules.

**Episodic Memory.** In cognitive psychology, Episodic Memory is another important part of long-term memory. It can store task-specific execution summaries (i.e., "autobiographical" records of past attempts). In RoboMemory, the Episodic Memory module is responsible for recording every interaction trajectory it has gone through. Including the sequence of actions the robot did and the feedback from the environment. The trajectory information can help to improve the planning ability of RoboMemory. For example, if a trajectory for completing a similar task is stored in episodic memory, it can guide the agent in completing the current task. As the agent only needs to follow the successful trajectory in the memory, it can reduce hallucinations or errors in the VLM planner. Technically, Episodic Memory shares the same storage and VLM-driven vector database update mechanism as Semantic Memory, ensuring consistent architectural design.

### 3.3 CLOSED-LOOP PLANNING MODULE FOR DYNAMIC ENVIRONMENT

The Closed-Loop Planning Module integrates information about the current task provided by the Spatial-Temporal Memory, Semantic and Episodic information recorded in long-term memory, and current observations to perform action planning. Each action is planned and passed on to the low-level executor for execution.

To enable closed-loop control in embodied environments, the Closed-Loop Planning Module adopts the Planner-Critic mechanism (Lei et al., 2025), which consists of the planner and the critic module. We denote the planner module as $\mathcal{P}$, while the critic module as $\mathcal{C}$. For each planning step, the planner generates a long-term plan consisting of multiple steps. However, due to the dynamics of embodied environments, the action sequence in the long-term plan may become outdated during the execution of the plan. Thus, before executing each step, we use the Critic model to evaluate whether

the proposed action in this step remains appropriate under the latest environment. If not, the planner will re-plan based on the latest information. The demonstration of this process is shown in Figure 2.

However, our experiments reveal that the original Planner-Critic mechanism may suffer from infinite loops. In the original mechanism, the first step of the action sequence output by the Planner is evaluated by the Critic before execution, which can lead to an infinite loop: if the Critic always demands replanning, no action will ever be executed. To address this, we modified the Planner-Critic mechanism so that the first step is not evaluated by the Critic. This ensures that even if the Critic persistently demands replanning, the RoboMemory will still execute actions. The detailed algorithm is shown in Appendix B.

### 3.4 Low-level Executor

The RoboMemory framework is a two-layer hierarchical agent framework. This design enables RoboMemory to accomplish longer-term tasks in the real world. The upper layer is responsible only for high-level planning, while the Low-level Executor carries out the actions planned by the upper layer in the real environment.

We employ a LoRA-finetuned VLA model, $\pi_0$ (Hu et al., 2022; Black et al., 2024), to generate manipulation actions, and a SLAM-based navigation model for locomotion. The low-level executor then translates high-level actions planned by RoboMemory into concrete arm and chassis movements in the real world.

Table 1: Comparison of Success Rates (SR) and Goal Condition Success Rates (GC) across difficulty levels (Base/Long) on EB-ALFRED and EB-Habitat benchmarks. Values are reported in percentages (%).

| Method | Type | Average | | EB-ALFRED | | | | EB-Habitat | | | |
| | | | | Base | | Long | | Base | | Long | |
| | | SR | GC | SR | GC | SR | GC | SR | GC | SR | GC |
|---|---|---|---|---|---|---|---|---|---|---|---|
| *Single VLM-Agents* | | | | | | | | | | | |
| GPT-4o | | 67.0 | 74.9 | 64.0 | 74.0 | 54.0 | 62.5 | 86.0 | 90.7 | **64.0** | 72.2 |
| GPT-4o-mini | | 30.5 | 40.9 | 34.0 | 47.8 | 0.0 | 17.0 | 74.0 | 77.5 | 14.0 | 21.3 |
| Claude-3.7-Sonnet | | 68.5 | - | 68.0 | - | **70.0** | - | 90.0 | - | 46.0 | - |
| Claude-3.5-Sonnet | Closed-source | 69.5 | 71.8 | **72.0** | 72.0 | 52.0 | 54.5 | **96.0** | **97.5** | 58.0 | 63.3 |
| Gemini-1.5-Pro | | 68.0 | 73.3 | 70.0 | 74.3 | 58.0 | 65.0 | 92.0 | 92.5 | 52.0 | 61.2 |
| Gemini-2.0-flash | | 57.0 | 61.5 | 62.0 | 65.7 | 58.0 | 62.0 | 82.0 | 82.0 | 26.0 | 36.2 |
| Llama-3.2-90B-Vision-Ins | | 40.5 | 46.6 | 38.0 | 43.7 | 16.0 | 24.0 | 94.0 | 94.5 | 14.0 | 24.3 |
| InternVL2.5-78B | | 47.0 | 52.9 | 38.0 | 42.3 | 42.0 | 49.0 | 80.0 | 82.0 | 28.0 | 38.2 |
| InternVL2.5-38B | Open-source | 37.5 | 42.6 | 36.0 | 37.3 | 26.0 | 36.5 | 60.0 | 61.5 | 28.0 | 35.0 |
| InternVL3-78B | | 49.5 | - | 38.0 | - | 36.0 | - | 84.0 | - | 40.0 | - |
| Qwen2.5-VL-72B-Ins | | 44.0 | - | 50.0 | - | 34.0 | - | 74.0 | - | 18.0 | - |
| *VLM-Agent Frameworks* | | | | | | | | | | | |
| Voyager (Qwen2.5-VL-72B-Ins) | | 46.5 | 66.4 | 56.0 | 73.2 | 32.0 | 54.2 | 76.0 | 87.0 | 22.0 | 51.0 |
| Reflexion (Qwen2.5-VL-72B-Ins) | | 38.3 | 51.1 | 48.0 | 54.0 | 10.0 | 33.0 | 80.0 | 84.2 | 15.0 | 33.0 |
| Cradle (Qwen2.5-VL-72B-Ins) | Baselines | 44.5 | 57.0 | 54.0 | 67.9 | 32.0 | 41.0 | 62.0 | 67.0 | 30.0 | 52.1 |
| RoboOS (Qwen2.5-VL-72B-Ins) | | 25.5 | 33.0 | 32.0 | 38.4 | 12.0 | 17.6 | 38.0 | 47.8 | 20.0 | 28.2 |
| RoboOS (RoboBrain2-32B) | | 20.0 | 25.7 | 32.0 | 37.2 | 8.0 | 13.2 | 28.0 | 34.8 | 12.0 | 17.4 |
| **RoboMemory (Qwen2.5-VL-72B-Ins)** | **Ours** | **70.5** | **79.7** | 68.0 | **75.5** | 66.0 | **81.3** | 86.0 | 88.0 | 62.0 | **74.0** |

## 4 Experiments

### 4.1 Benchmarks

To evaluate RoboMemory's task planning ability, we select a subset of the EmbodiedBench EB-ALFRED and EB-Habitat benchmark (Yang et al., 2025a). We selected the Base and Long subsets because they aim to test the agent's planning ability. The Base and Long subsets of the two benchmarks comprise 200 tasks for complex embodied tasks. The EB-ALFRED and EB-Habitat benchmarks provide a visually grounded operational setting that closely mimics real-world conditions (see Appendix F for environment details), enabling direct comparison with established baselines.

Moreover, we set up an environment to test the interactive environmental learning ability of RoboMemory in the real world.

## 4.2 SETTINGS & BASELINES

To facilitate comparisons, we consider two types of baselines. First, we choose the advanced closed-source and open-source VLMs as a single agent. We compare their performance with RoboMemory. For closed source VLMs, we choose GPT-4o and GPT-4o-mini (OpenAI, 2024; Hurst et al., 2024), Claude3.5-Sonnet and Claude-3.7-Sonnet (Anthropic, 2024), Gemini-1.5-Pro and Gemini-2.0-flash (Team et al., 2024; DeepMind, 2024). For open source VLMs, we choose Llama-3.2-90B-Vision-Ins (Meta, 2024), InternVL-2.5-78B/28B (Chen et al., 2024b), InternVL-3-72B (Zhu et al., 2025), and Qwen2.5-VL-72B-Ins (Bai et al., 2025). Secondly, we choose three agent frameworks: (1) Reflexion (Shinn et al., 2024), which introduces a simple long-term memory and a self-reflection module. Reflexion uses the self-reflection module to summarize experiences as long-term memory, thereby enhancing the model's capabilities. (2) Voyager (Wang et al., 2023), which utilizes a skill library as its procedural memory, is a widely used baseline for embodied agent planning. (3) Cradle (Tan et al., 2024), which proposes a general agent framework with episodic and procedural memory and gains good performances at various multi-model agent tasks. (4) RoboOS (Tan et al., 2025), which proposes an embodied agent framework that consists of Scene-Graph based Spatial Memory.

In our experiments, each agent framework is tested using Qwen2.5-VL-72b-Ins (Team, 2024) with temperature set as 0. For the RoboOS framework, we test it on RoboBrain2-32B (Team et al., 2025), where the RoboBrain2-32B model is designed for the RoboOS framework.

The Qwen2.5-VL-72b-Ins represents a high-performing open-source alternative. Notably, the Qwen2.5-VL-72b-Ins demonstrates performance comparable to advanced closed-source VLMs in several benchmark tasks (White et al., 2024). We use the Qwen3-Embedding model (Zhang et al., 2025) to create embedding vectors for RAGs in RoboMemory. For the Low-level Executor, since EB-ALFRED provides high-level action APIs, we use the low-level executor provided by Embodied-Bench instead of the VLA-based method.

We define two evaluation metrics to assess the performance: (1) Success Rate (SR), which is the ratio of completed tasks to the total number of tasks in each difficulty level. This metric reflects the agent's ability to complete tasks across randomly generated scenarios. (2) Goal Condition Success Rate (GC), which is the ratio of intermediate conditions achieved to the maximum possible score in each scenario. An GC of 100% indicates that the task is completed in the given scenario. These two metrics can be computed as:

$$SR = \mathbb{E}_{x \in \mathcal{X}} \left[ \mathbb{1}_{SCN_x = GCN_x} \right] \tag{3}$$

$$GC = \mathbb{E}_{x \in \mathcal{X}} \left[ \frac{SCN_x}{GCN_x} \right] \tag{4}$$

Where $\mathcal{X}$ denotes the test subset, and $x$ represents a test task. The success condition number ($SCN_x$) refers to the number of conditions the agent has accomplished, while the global condition number ($GCN_x$) indicates the total number of conditions required for task completion. The task is considered successful if $SCN_x = GCN_x$.

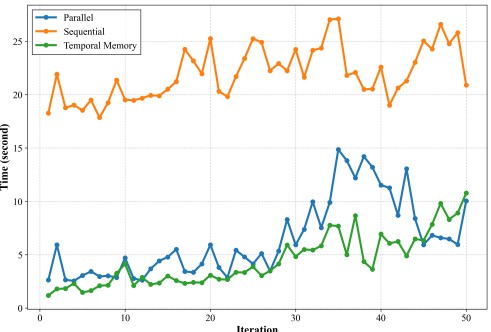

Figure 3: Efficiency improvement of Comprehensive Embodied Memory System

Table 2: Ablation Study on RoboMemory's Success Rate (SR)

| Method | Avg. | Base | Long |
|---|---|---|---|
| **RoboMemory** | **67%** | **68%** | **66%** |
| - w/o critic | 55 % | 60 % | 50% |
| - w/o spatial memory | 47 % | 52 % | 42 % |
| - w/o long-term memory | 57% | 66% | 48% |
| - w/o episodic memory | 62% | 68% | 56% |
| - w/o semantic memory | 58% | 66% | 50% |

## 4.3 MAIN RESULTS

As shown in Table 1, our model achieves significant improvements over both single VLM agents and Agent frameworks on the EB-ALFRED and EB-Habitat. Compared to the SOTA Single VLM-Agent model, Claude3.5-Sonnet, RoboMemory with Qwen2.5-VL-72B-Ins backbone improves the average SR by 1% and GC by 7.9%. This demonstrates RoboMemory's superiority over single VLM-Agents, proving that an Agent framework with open-source models can outperform closed-source SOTA models. Furthermore, when tested against other VLM-Agent frameworks, RoboMemory also shows substantial gains. This is because, unlike other frameworks, RoboMemory's brain-like memory system provides embodied models with more accurate and persistent contextual information. Additionally, the Planner-Critic mechanism provides a closed-loop planning ability, which helps the RoboMemory gain better performance in long-term tasks. Because the RoboMemory can detect and try to overcome possible failures. And it is more robust when encountering unexpected situations.

## 4.4 EFFICIENCY ANALYSIS

To evaluate the efficiency of the Comprehensive Embodied Memory module, we tasked RoboMemory with executing 10 long-horizon tasks, each comprising approximately 50 steps. We exclusively measured the wall-clock time consumed by memory update and retrieval operations. We analyzed the scaling behavior of memory update latency across three distinct configurations: (1) fully parallel update and retrieval across all memory modules; (2) sequential update of each memory module without parallelism; and (3) update of only the most fundamental memory component: the Temporal Memory. Results are presented in Figure 3. As shown, our parallel update strategy enables updating a multi-module memory system with latency comparable to that of updating a single base memory module. This demonstrates the critical efficiency gains afforded by parallelization across the memory architecture.

## 4.5 ABLATION STUDIES

We used the full Base and Long Subset from EB-ALFRED to validate RoboMemory's effectiveness. We removed each component systematically and observed performance changes across task categories. We use the success rate as our metric. Results are shown in Table 2.

**Long-term Memory** Adding long-term memory significantly improved RoboMemory's success rate. The experiment shows that it enables interactive environmental learning while attempting to complete tasks. The semantic memory learns low-level skills' properties, such as in what circumstances an action may fail. The temporal memory records all task attempts (successful/failed), providing valuable experience at the task level and giving insight into how to complete a task successfully. This helps the RoboMemory predict action outcomes and avoid ineffective attempts. This ability indicates that the RoboMemory has an interactive environmental learning capability.

**Spatial Memory** Spatial memory is crucial for embodied agents, especially given that current pretrained VLMs have limited spatial understanding ability. Our novel dynamic KG update algorithm enables KG-based spatial memory in dynamic environments. This spatial reasoning helps RoboMemory handle partially observable embodied settings.

**Critic Module** Table 2 shows performance without the critic module (55% vs 67% with full system). This drop highlights how the critic's closed-loop planning adapts to dynamic environments. It helps RoboMemory recover from failures faster and handle unexpected situations better.

## 4.6 REAL-WORLD ROBOT DEPLOYMENT

To evaluate RoboMemory's interactive environmental learning capability in the real world, we designed a kitchen environment inspired by EB-ALFRED and EB-Habitat. The scene contains 5 navigable points, 8 interactive objects, and over 10 non-interactive (but potentially distracting) items. The environment is shown in Figure 4. In the real world, we use interactive environmental video recordings captured during action execution (rather than static snapshots taken after action completion) as RoboMemory's input. This provides a more temporally coherent perception. We created three task categories (5 tasks each). Tasks are matched to EB-ALFRED's Base subset (avg. oracle: 10–20 steps), though actual executions often exceed 20 steps due to search and error recovery.

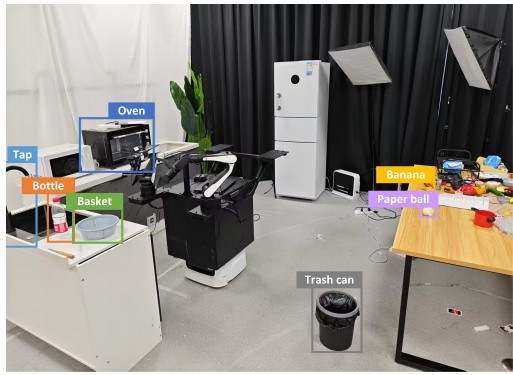

Figure 4: Visualization of the experimental environment.

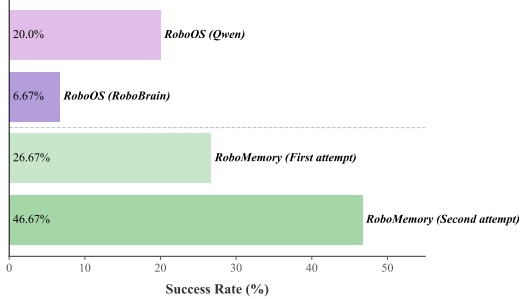

Figure 5: Real-world experiment results. "Qwen" denotes Qwen2.5-VL-72B-Ins; "RoboBrain" denotes RoboBrain2.0-32B.

Due to search and error recovery, the robot often exceeds 20 steps per task. Additional hardware experiment details are in Appendix F.

To test the interactive environmental learning ability of RoboMemory, we run each task twice without clearing long-term memory between attempts. Meanwhile, We compared RoboMemory against RoboOS as baselines. The success rates for first and second attempts and different settings of RoboOS are shown in Figure 5.

The second attempt showed significantly higher success rates. This proves RoboMemory's long-term memory effectively guides subsequent tasks in real embodied environments. Key observations include: (1) Closed-loop error recovery: RoboMemory retries failed actions when possible, even if the low-level executor (VLA model) fails. (2) Spatial reasoning: RoboMemory remembers object locations and spatial relationships using its memory. (3) Interactive environmental learning: RoboMemory analyzes failure causes reasonably. These analyses guide future decisions. Detailed examples demonstrating these capabilities and further discussions are provided in Appendix G.

Moreover, we observed a significant drop in task success rates when deploying the agent with the Low-level Executor in real-world environments. This performance degradation primarily stems from the executor's inherent limitations: (1) The VLA model exhibits unreliable instruction-following capabilities, frequently failing during grasping actions or selecting incorrect objects; (2) Pre-trained VLM models demonstrate inadequate video understanding - while capable of recognizing static objects, they struggle to interpret dynamic visual information such as action failures or state changes. These limitations collectively contribute to the reduced performance compared to simulated environments.

## 5 CONCLUSION AND FUTURE WORK

In summary, RoboMemory, a brain-inspired multi-memory framework, facilitates long-horizon planning and interactive environmental learning in real-world embodied systems by addressing key challenges such as memory latency, task correlation capture, and planning loops. Experiments on EmbodiedBench demonstrate that RoboMemory outperforms state-of-the-art closed-source VLMs and agent frameworks, with ablation studies confirming the critical roles of the Critic module and spatial/long-term memory. Real-world deployment further validates its interactive learning capability through improved success rates in repeated tasks. Despite limitations arising from reasoning errors and executor dependence, RoboMemory provides a foundation for generalizable, memory-augmented agents, with future work aimed at refining reasoning and enhancing execution robustness.

A notable open challenge in hierarchical embodied agents, including RoboMemory, lies in the interface between high-level planners and low-level executors. Existing frameworks typically rely on language instructions to convey actions, yet some execution details (e.g., precise grasp points) are difficult to describe textually and are better captured through other modalities, such as vision. While our current work emphasizes long-term planning and interactive learning, future research may improve generalization by developing richer multimodal interactions between the agent and executor.

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

## A    STATEMENT OF LLM USAGE

In this article, the LLM participated in the following tasks: (1) assisting in revising and polishing the manuscript, and (2) serving as an experimental subject in various experiments.

## B    ADDITIONAL ALGORITHMS

---

**Algorithm 1** RoboMemory Execution Process

---

**Require:** Task description $\mathcal{T}$, Initial observation $\mathcal{O}_0$, Max steps $T_{max}$
**Require:** Modules: Step Summarizer $\mathcal{S}$, Query Generator $\mathcal{Q}$; Memory $\mathcal{U}, \mathcal{R}$; Planner $\mathcal{P}$; Critic $\mathcal{C}$; Executor $\mathcal{E}$
  1: **Initialize:** Global step $t \leftarrow 0$, Memory $M_t \leftarrow \emptyset$
  2: **Initial Perception:**
  3:   $s_t, q_t \leftarrow \mathcal{SQ}(\mathcal{O}_t)$ {call the step summarizer and query generator in parallel}
  4:   $M_t \leftarrow \mathcal{U}(M_t, s_t)$ {Initialize Memory with first observation}
  5: **while** $t < T_{max}$ **and** Task $\mathcal{T}$ not completed **do**
  6:     **Retrieval Phase:**
  7:     $r_t \leftarrow \mathcal{R}(M_t, q_t)$ {Parallel retrieval from $L$ memory modules}
  8:     **Planning Phase:**
  9:     $\mathbf{A} \leftarrow \mathcal{P}(r_t, \mathcal{O}_t, \mathcal{T})$ {Generate action sequence $\mathbf{A} = [a_1, a_2, \ldots, a_K]$}
10:     **Execution Phase (Closed-Loop):**
11:     **for** $k = 1$ **to** $|\mathbf{A}|$ **do**
12:       Let $a_k$ be the current action to execute
13:       *execute_flag* $\leftarrow$ False
14:       **if** $k = 1$ **then**
15:         *execute_flag* $\leftarrow$ True {Skip Critic for the first step to avoid infinite loops}
16:       **else**
17:         {Re-evaluate context for subsequent steps}
18:         $r_{curr} \leftarrow \mathcal{R}(M_t, q_{curr})$
19:         **if** $\mathcal{C}(a_k, r_{curr}, \mathcal{O}_t, \mathcal{T})$ is **True then**
20:           *execute_flag* $\leftarrow$ True
21:         **else**
22:           **break** {Critic rejects action; trigger re-planning}
23:         **end if**
24:       **end if**
25:       **if** *execute_flag* is **True then**
26:         $\mathcal{O}_{t+1} \leftarrow \mathcal{E}(a_k)$ {Execute action via low-level executors}
27:         $t \leftarrow t + 1$
28:         **Memory Update (Perception → Memory):** $s_t, q_t \leftarrow \mathcal{SQ}(\mathcal{O}_t)$ {Generate query and summary in parallel}
29:         $M_t \leftarrow \mathcal{U}(M_{t-1}, s_t)$ {Parallel update of all modules}
30:       **end if**
31:     **end for**
32: **end while**

---

## C    ADDITIONAL EXPERIMENTS

### C.1    ERROR ANALYSIS

We summarize the common errors of RoboMemory in the previous experiments. We classify errors into three main types: planning errors, reasoning errors, and perception errors.

The planning errors occur when the planner fails to generate correct actions. The reasoning errors occur when the planner and critic cannot properly process input information (including current observations and memory), even when the input is correct. Perception errors occur when incorrect information is provided to the planner-critic module.

We analyze RoboMemory trajectories for failed tasks. We identify error types based on the above definitions. A single task may contain multiple errors. We calculate the occurrence probability of each error type to show RoboMemory's strengths and weaknesses. The results are shown in Figure 6.

We can observe that among all error types, the planning errors are the most common. This means that even though the memory modules can provide comprehensive information about the RoboMemory agent's previous experience and spatial and temporal memory for the current task, the planner module may still not provide good action plans. This may be due to the capability of the pretrained base model.

The most common perception error is the hallucination error. We can observe that although some hallucinations can be handled by the critic

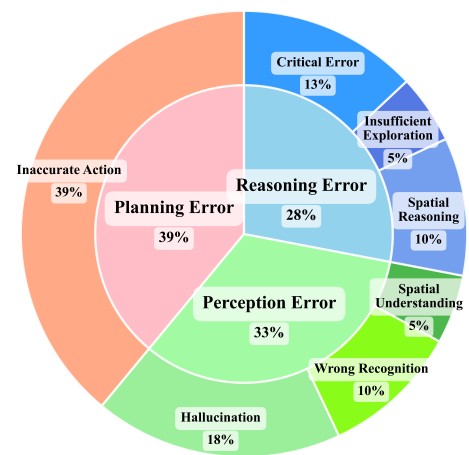

Figure 6: The reason why RoboMemory failed to complete the task.

module or memory information, there are still some cases in which the planner ignores all insights from memory and critic and fails to complete the task.

The detailed examples and discussions are provided in Appendix G.

### C.2 ADDITIONAL EFFICIENCY ANALYSIS

We analyze the evolution of the spatial KG during long trajectories in EB-ALFRED, focusing on the first 20 iterations (with 95% confidence intervals). As shown in Figure 7, the total number of spatial relationships in the KG (red line) increases gradually over iterations as RoboMemory is exploring the environment. In contrast, the number of relationships retrieved for update at each iteration (blue line) remains relatively stable, typically ranging around 10 edges per iteration. This stability is achieved because our method only updates a local subgraph relevant to the current observation.

We define the retrieval ratio as the proportion of relationships updated at each iteration relative to the total number of relationships in the KG. As shown in Figure 7, this ratio (illustrated by gray bars) decreases steadily from 76% initially to 28% at iteration 20. This trend indicates that, as the KG grows, each update affects a progressively smaller fraction of the entire graph. This demonstrates that our spatial KG update mechanism effectively localizes modifications, ensuring computational efficiency and mitigating interference through context-aware incremental updates.

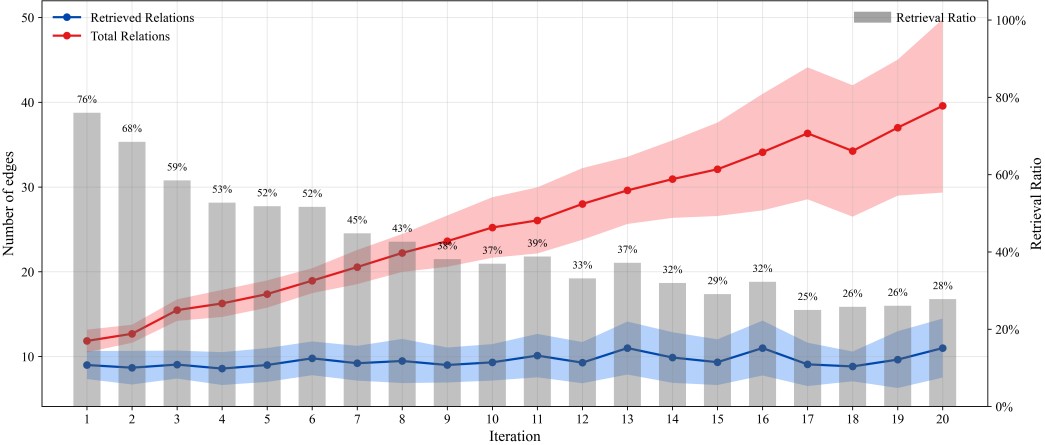

Figure 7: Average relationships related to update in spatial memory in each step.

# D ADDITIONAL RELATED WORK

Table 3: Comparison of Memory-Related Methods in Embodied Agent

| Method | Multimodal | Episodic | Semantic | Spatial | Temporal | Procedural | Memory Implementation | Real Robot |
|---|---|---|---|---|---|---|---|---|
| NeSyC (Choi et al., 2025) | ✓ | | ✓ | | ✓ | | Symbolic logic rules | ✓ |
| Reflexion (Shinn et al., 2024) | | | ✓ | | ✓ | | Buffer | |
| Voyager (Wang et al., 2023) | | | | | | ✓ | RAG | |
| MSI-Agent (Fu et al., 2024a) | | | ✓ | | ✓ | | Database, RAG | |
| CoELA (Zhang et al., 2023) | ✓ | ✓ | ✓ | | | ✓ | Top-down semantic map | |
| Cradle (Tan et al., 2024) | ✓ | ✓ | | | ✓ | ✓ | RAG | |
| Agent-S (Agashe et al., 2024) | ✓ | ✓ | ✓ | | ✓ | | RAG | |
| Expel (Zhao et al., 2024) | | | ✓ | | ✓ | | Buffer | |
| AutoManual (Chen et al., 2024a) | | ✓ | ✓ | | | ✓ | Buffer | |
| HiRobot (Shi et al., 2025) | ✓ | | | | | | / | ✓ |
| Being-0 (Yuan et al., 2025) | ✓ | ✓ | | | ✓ | | Buffer | ✓ |
| RoboOS (Tan et al., 2025) | ✓ | | | ✓ | ✓ | | Scene graph, database | ✓ |
| RoboMemory (Ours) | ✓ | ✓ | ✓ | ✓ | ✓ | | RAG,KG | ✓ |

# E DYNAMIC SPATIAL MEMORY UPDATE ALGORITHM

## E.1 DETAILED ALGORITHM OF SPATIAL KG UPDATE

---

**Algorithm 2 Retrieval-based Incremental Knowledge Graph Update Algorithm**

---

**Require:** New spatial knowledge graph $G_{\text{new}} = (V_{\text{new}}, E_{\text{new}})$, main spatial knowledge graph $G = (V, E)$, queries $q \in Q$, entity & query embeddings $\mathcal{E} : V \cup Q \to \mathbb{R}^d$, maximum number of retrieved vertices $n$, maximum k hops $k$, vlm-base conflict resolver ResolveConflict$(\cdot)$

**Ensure:** Updated consistent knowledge graph $G'$

1: $V_{\text{similar}} \leftarrow \bigcup_{q \in Q} \text{TopK}_n (\{v \in V \mid \text{cosine\_sim}(\mathcal{E}(q), \mathcal{E}(v))\})$ {For each query entity $q$, retrieve its top-$n$ most similar vertices in $G$ by cosine similarity of embeddings; take the union over all $q \in Q$.}

2: $V_{\text{expand}} \leftarrow \text{K-hop}_k(V_{\text{similar}}, G)$ {all nodes within $k$ hops from any node in $V_{\text{similar}}$}

3: $V_{\text{retrieved}} \leftarrow V_{\text{similar}} \cup V_{\text{expand}}$

4: $V_{\text{merged}} \leftarrow V_{\text{retrieved}} \cup V_{\text{new}}$

5: $G_{\text{union}} \leftarrow (V \cup V_{\text{new}}, E \cup E_{\text{new}})$ {Combine the main graph and new observations into a unified graph.}

6: $G_{\text{local}} \leftarrow \text{InducedSubgraph}(V_{\text{merged}}, G_{\text{union}})$ {Extract the subgraph induced by $V_{\text{merged}}$, containing all old and new edges among these nodes.}

7: $G_{\text{updated}} \leftarrow \text{ResolveConflict}(G_{\text{local}}, G_{\text{new}})$ {based on $G_{\text{new}}$, VLM update the relationship among different vertices in $G_{\text{local}}$}

8: $G' \leftarrow (G \setminus G_{\text{local}}) \cup G_{\text{updated}}$ {Replace the old subgraph in $G$ with the conflict-resolved updated subgraph.}

9: Remove isolated vertices from $G'$

10: **return** $G'$

---

Spatial Memory is a dynamically updated KG-based module designed to overcome agents' limitations in spatial reasoning. Specifically, our Spatial Memory is formulated as a directed KG $G = (V, E)$, where $V$ denotes the set of all objects in the environment. Each object is a vertex in the KG. Each object's name is encoded into a single semantic embedding vector via a pretrained embedding model (Zhang et al., 2025). The edge set $E$ captures spatial relationships between objects, each represented as a triple (e.g., $[obj_1, \text{relationship}, obj_2]$). To update the Spatial KG, we need to continuously extract new relationships from current observations and update the old relationships in the Spatial KG. We use a VLM-based conflict resolver to address this problem. However, the more relationships provided to the conflict resolver, the more time it needs to update the KG. So we need to update relationships that are only related to the current situation. We design an algorithm that retrieves a sub-graph of KG that includes all vertices related to the current situation and both old and new relationships among them. We provide the sub-graph and new relationships to the conflict resolver. We need the conflict resolver to update the sub-graph based on information from the new relationships. The algorithm is shown in Algorithm 2.

To update $G$, we make use of the information provided by the step summarizer and query generator from the information preprocessor introduced in Section 3.1. First, a pretrained VLM-based Relation Retriever extracts the latest spatial relationships $G_{\text{new}} = (V_{\text{new}}, E_{\text{new}})$ from the information provided by the step summarizer, which records high-level information in the current observation. Next, natural language queries (represented as $q \in Q$) (provided by the query generator) are used to retrieve relevant object vertices from $G$ via cosine similarity search. We select the top $n$ similar vertices compared with the query. These vertices are represent by $V_{\text{similar}}$.

Spatial KG maintains the relationships among different objects, so if we want to retrieve spatial information from spatial KG, we need to search for other objects that are related to the objects we observed in the current observation. In this way, we not only remember objects we can see, but also know the spatial information of the objects we cannot see. So we choose the k-hop algorithm to expand $V_{\text{similar}}$ using a K-hop neighborhood algorithm to capture contextually related objects. The K-hop algorithm is represented as K-hop$_k(V, G)$, which returns all vertices reachable within $\leq k$ hops from any vertices in $V$. The retrieved vertices are $V_{\text{expand}}$. We combine the vertices retrieved by cosine similarity and their K-hop neighbors to $V_{\text{retrieved}}$.

However, we need to resolve the conflict between new and old relationships. So $V_{\text{retrieved}}$ and the relationships among $V_{\text{retrieved}}$ is not enough. We need new vertices and relationships involved in the graph we provided to the VLM-based conflict resolver. To extract all vertices and relationships for the VLM-based conflict resolver and relationships, we not only need the relationships from KG (old information) and $G_{\text{new}}$, which represent new information. We need to connect old information and new information. To achieve this goal, we merge $G_{\text{new}}$ to $G$, which add new edges and vertices to $G$. We denote the merged KG as $G_{\text{union}}$. In $G_{\text{union}}$, we mix out-of-date and latest information. Then, we extract an induced sub-graph of $V_{\text{merged}} = V_{\text{retrieved}} \cup V_{\text{new}}$ from $G_{\text{union}}$. As both vertices from old graph $G$ and new graph is mixed in $V_{\text{merged}}$ and both edges from old and new graph is in $G_{\text{union}}$, the retrieved induced sub-graph $G_{\text{local}}$ contains all out-of-date and latest relationships among vertices that is related to current situation.

As $G_{\text{local}}$ contains all out-of-date and latest relationships and those relationships may have conflicts, a VLM-based conflict resolver (represent as ResolveConflict($\cdot$)) is designed to resolve conflicts in $G_{\text{local}}$, and make sure that the relationship is the latest. The conflict resolver will take in $G_{\text{new}}$ and $G_{\text{local}}$, where $G_{\text{local}}$ is the graph waiting for update and $G_{\text{new}}$ provide update signal. The VLM-based conflict resolver will perform necessary updates such as adding vertices or inserting, deleting, or modifying relationships in $G_{\text{local}}$ based on $G_{\text{new}}$. The reconciled subgraph is then merged back into $G$, and any vertices that have lost all connections to other vertices during the update are pruned.

This design offers two key advantages: (1) Efficiency via localized updates: By restricting modifications to a context-relevant subgraph, we significantly reduce the number of relationships processed per update. Since VLMs struggle with reasoning over large sets of relationships, this constraint substantially improves both the efficiency and effectiveness of VLM-based KG updates. (2) Dynamic adaptability: The system continuously maintains up-to-date spatial knowledge, enabling agents to operate robustly in dynamic real-world environments.

### E.2 Example of Dynamic Spatial Memory Update process

In RoboMemory's Spatial Memory, the KG is dynamically constructed during environment exploration. As illustrated in Figure 8, we demonstrate the progressive expansion of the KG in Spatial Memory as the agent navigates through the environment. The figure indicates a continuous growth in the number of both vertices and edges of the KG as exploration progresses.

Notably, the KG undergoes dynamic updates through RoboMemory's environmental interactions. For example, the initial KG state displays the relation "I am near the apple. But as the agent picks up the apple in the third step, in the fourth KG, the relationship becomes "I hold the apple". This demonstrates RoboMemory's capability for dynamic KG maintenance and expansion.

By querying this KG, the Planner-Critic module gains access to rich spatial information, empowering RoboMemory with robust spatial memory capabilities that significantly enhance its performance in both TextWorld and EmbodiedBench environments.

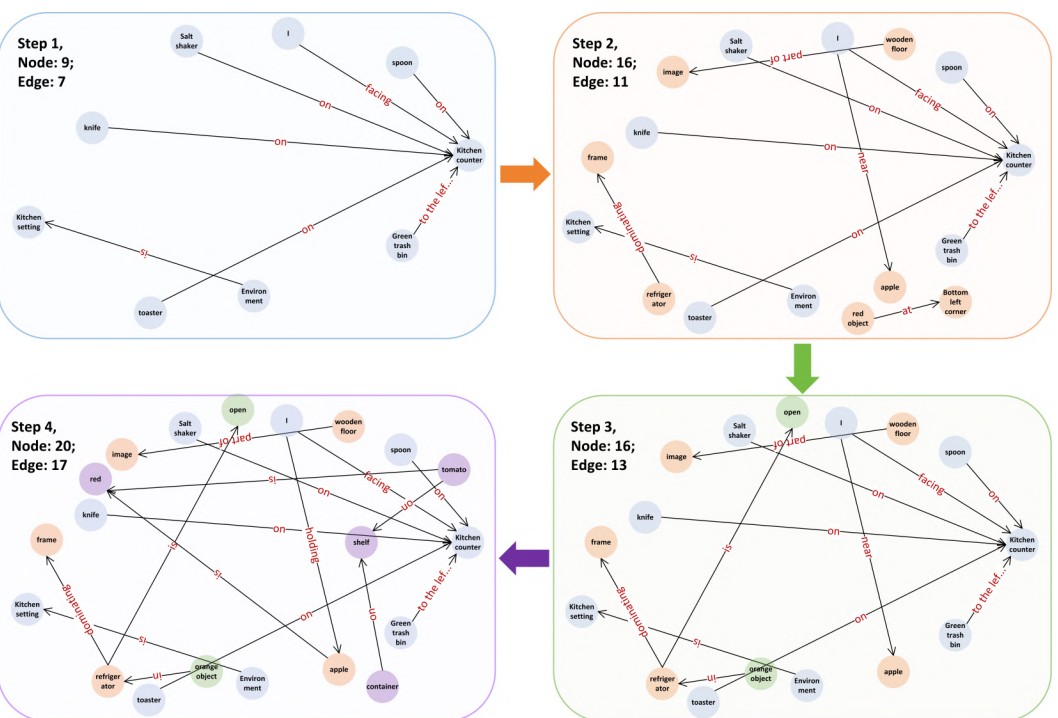

Figure 8: Visualization of Spatial Memory's dynamic update process.

### E.3 PROOF OF DYNAMIC SPATIAL MEMORY UPDATE ALGORITHM

**Theorem 1** (Upper Bound on K-hop Vertex Extraction in Directed Graphs). *Let $G = (V, E)$ be a finite directed graph with maximum out-degree $D \geq 1$, and let $\mathcal{S} \subseteq V$ be a set of $M$ source vertices. Define the K-hop neighborhood $\mathcal{N}_K(s)$ of a vertex $s \in \mathcal{S}$ as the set of vertices reachable from $s$ via directed paths of length at most $K$. Then the total number of distinct vertices in the union of all K-hop neighborhoods,*

$$\mathcal{N}_K(\mathcal{S}) = \bigcup_{s \in \mathcal{S}} \mathcal{N}_K(s),$$

*Satisfies the following upper bound:*

$$|\mathcal{N}_K(\mathcal{S})| \leq \begin{cases} M \cdot \dfrac{D^{K+1} - 1}{D - 1}, & \text{if } D > 1, \\ M \cdot (K + 1), & \text{if } D = 1. \end{cases}$$

*Proof.* For any vertex $s \in \mathcal{S}$, the number of distinct vertices reachable from $s$ within $i$ hops is at most $D^i$, assuming the worst-case scenario where each vertex encountered has the maximum out-degree $D$, and all neighbors are distinct and non-overlapping.

Thus, the size of the K-hop neighborhood of a single vertex satisfies:

$$|\mathcal{N}_K(s)| \leq \sum_{i=0}^{K} D^i = \begin{cases} \dfrac{D^{K+1} - 1}{D - 1}, & \text{if } D > 1, \\ K + 1, & \text{if } D = 1. \end{cases}$$

Since there are $M$ such source vertices and assuming no overlaps between their K-hop neighborhoods (worst case), the union size satisfies:

$$|\mathcal{N}_K(\mathcal{S})| \leq M \cdot |\mathcal{N}_K(s)|.$$

Substituting the bound on $|\mathcal{N}_K(s)|$ gives the result. □

**Theorem 2** (Upper Bound for K-hop Vertex Extraction in Normalized Directed Graphs). *Let $G = (V, E)$ be a finite directed graph with $|V| = n$ vertices. Assume the maximum out-degree is at most $D_{\max} = Dn$, and the maximum in-degree is at most $N_{\max} = Nn$, where $D, N \in (0, 1]$ are constants. Let $\mathcal{S} \subseteq V$ be a set of $M$ source vertices. Define $\mathcal{N}_K(\mathcal{S})$ as the union of all vertices reachable from $\mathcal{S}$ via paths of length at most $K$, using only outgoing edges. Then the number of extracted vertices satisfies:*

$$|\mathcal{N}_K(\mathcal{S})| \leq \min\left\{n, \ M \cdot \frac{(Dn)^{K+1} - 1}{Dn - 1}\right\}.$$

*In particular, when $Dn \gg 1$, we have the approximation:*

$$|\mathcal{N}_K(\mathcal{S})| \lessapprox M \cdot (Dn)^K.$$

*Proof.* For each vertex $s \in \mathcal{S}$, the maximum number of reachable vertices within $i$-hops is at most $(Dn)^i$ under the assumption of maximum out-degree and no overlap.

Summing over hops from 0 to $K$, we get for each root:

$$|\mathcal{N}_K(s)| \leq \sum_{i=0}^{K} (Dn)^i = \frac{(Dn)^{K+1} - 1}{Dn - 1}.$$

Assuming no overlap among the $M$ source vertex expansions (worst case), we have:

$$|\mathcal{N}_K(\mathcal{S})| \leq M \cdot \frac{(Dn)^{K+1} - 1}{Dn - 1}.$$

Since the total number of vertices in the graph is $n$, this quantity is also trivially bounded above by $n$, yielding the result. $\square$

# F ADDITIONAL ENVIRONMENT SETTINGS

## F.1 EB-ALFRED AND EB-HABITAT

We adopt the same environment parameters as in EmbodiedBench. The maximum steps per task are set to 30, with image inputs of size $500 \times 500$. The temporal memory buffer length is set to 3. However, we modified the action formats of EB-ALFRED and EB-Habitat to simulate real-world scenarios better. Specifically, we define different action APIs (Python functions), where each action takes an object parameter indicating its target. We extract all possible objects from the environment as inputs to the Agent. The Agent must select appropriate actions and object parameters based on task requirements. Compared to the original interaction method in EmbodiedBench (which enumerates all possible actions, including both action names and target objects, and requires the Agent to choose), our approach offers greater flexibility. The detailed action APIs are presented in Table 4.

Since EB-ALFRED and EB-Habitat provide comprehensive high-level action APIs, we do not employ the VLA-Based Low-Level Executor in these environments. Instead, we utilize the built-in low-level controllers from EmbodiedBench.

## F.2 ADDITIONAL SETTINGS FOR BASELINES

**EB-ALFRED.** For our single VLM-Agent baseline, we utilized the reported results from EmbodiedBench paper to establish a consistent benchmark, where the agent relies on a basic interaction history as its memory module. For other VLM frameworks, we replicated the experimental setups as described in both EmbodiedBench and the respective original papers. Crucially, to familiarize all baseline agents with the EmbodiedBench environment, we supplemented them with a few-shot example and a comprehensive catalog of actionable objects—applying the exact same conditions as those used for the single VLM-Agent benchmark.

Table 4: Robot Action Command For different environments

| Action Type | EB-ALFRED | EB-Habitat | Real World |
|---|---|---|---|
| Navigation | find(obj) | navigate(point) | navigate_to(point) |
| Pick Up Object | pick_up(obj) | pick(obj) | pick_up(obj) |
| Drop to Ground | drop() | − | − |
| Place to Receptacle | put_down() | place(rec) | put_down_to(rec) |
| Open Object | open(obj) | open(obj) | open(obj) |
| Close Object | close(obj) | close(obj) | close(obj) |
| Turn On | turn_on(obj) | − | turn_on(obj) |
| Turn Off | turn_off(obj) | − | turn_off(obj) |
| Slice Object | slice(obj) | − | − |
| Task Complete | − | − | task_complete() |

### F.3 REAL-WORLD EXPERIMENTS

We construct a common kitchen scenario to evaluate the RoboMemory framework's interactive environmental learning capabilities in real-world settings. Using Mobile ALOHA (Fu et al., 2024b) as our physical robotic platform, we design three categories of tasks: (1) Pick up & put down: The agent must locate a specified object among all possible positions and place it at a designated location. This task tests the model's basic object-searching and planning abilities. (2) Pick up, operate & put down: Building upon the first task, the agent must additionally perform operations such as heating or cleaning the object. This task requires longer-term planning, which is crucial in embodied environments. (3) Pick up, gather & put down: The agent must place specified objects into a movable container and then move the container to a target location. This task evaluates the agent's understanding of object relationships, requiring it to remember the positions of at least two objects (the container and the target item) and their spatial relationship. For each type of task, we design 5 tasks. So our experiments include 15 long-term real-world tasks.

To adapt to the real-world setup, we define high-level action APIs similar to those in EB-ALFRED and EB-Habitat. Additionally, we train a VLA-based model to execute tasks according to our action APIs. The detailed action APIs are presented in Table 4.

For the low-level executor, we use one main camera and two arm-mounted cameras as input, each with a resolution of $640 \times 480$. The temporal memory buffer length is set to 3.

In our experiments, we set the maximum steps per task to 25. We also provide an API for actively terminating tasks. Since real-world environments lack direct success/failure feedback, RoboMemory must autonomously determine task completion. To prevent excessively long task execution, we enforce termination after 25 steps if no success is achieved. A single main camera ($640 \times 480$ resolution) records video during action execution as input for RoboMemory's higher-level processing.

Table 5: Dataset statistics and training hyperparameters for robotic manipulation tasks.

| Dataset Statistics | | Training Configuration | |
|---|---|---|---|
| **Action Type** | **#Episodes** | **Parameter** | **Value** |
| Turning on/off faucet | 142 | Optimizer | AdamW |
| Picking up & Placing basket on counter | 63 | Batch size | $32 \times 6$ |
| Picking up & Placing basket in sink | 72 | Training steps | 10,000 |
| Picking up & Placing banana into basket | 114 | Learning rate | $6.12 \times 10^{-5}$ |
| Throwing bottle into trash bin | 132 | warm up step | 500 |
| Placing gum box on dish | 120 | **LoRA Configuration** | |
| Picking up & Placing cup on plate | 51 | rank | 16 |
| Picking up & Placing dish into sink | 69 | $\alpha$ | 16 |
| Throwing paper ball into trash bin | 135 | **Resource Usage** | |
| Open/close oven | 142 | GPU | A100-80GB $\times 6$ |
| **Total episodes** | **1040** | Training time | 12 hours |

### F.4 Training Details of Low-Level Executor

We use the $\pi_0$ model as our foundation model. We collected 1,040 data samples over 10 types of tasks for fine-tuning. We use LoRA fine-tuning to save resources during fine-tuning. The specific fine-tuning parameters and action types are given in Table 5. For tasks involving both pick-up and place actions, we split these tasks into separate pick-up and place actions. These are then treated as two distinct data samples during training. The separation of pick-up and place action allows the VLA to carry an object in its hand. For training, we used a server with six A100-80GB GPUs. The total training time was 12 hours.

Besides, we use the built-in LiDAR SLAM system of the Mobile ALOHA robot base as the navigation action actuator. We define five typical navigation points, similar to EB-Habitat. We used SLAM to navigate between these navigation points.

### F.5 Hyperparameters of RoboMemory

In this section, we describe the hyperparameter settings for the upper brain of RoboMemory: the information preprocessor and the Comprehensive Embodied Memory. Importantly, we use a unified set of hyperparameters across all experimental settings, including EB-ALFRED, EB-Habitat, and real-world deployments.

The information preprocessor consists of two components: a step summarizer and a query generator. Given multimodal inputs at each step, the step summarizer produces a single natural language description, while the query generator concurrently formulates $4-5$ distinct natural language queries.

The Comprehensive Embodied Memory integrates four memory modules: Temporal, Spatial, Semantic, and Episodic Memory. The Temporal Memory is implemented as a fixed-size buffer with a maximum capacity of $4$ entries. For Spatial Memory, during similarity-based retrieval, we first identify the top $N = 3$ most relevant vertices and then perform a K-hop graph traversal with $K = 2$. The Episodic Memory retrieves the top $N = 5$ most relevant past experiences for each query. The Semantic Memory maintains hierarchical summaries at both the action and task levels; during retrieval, it returns $N_s = 2$ action-level and $N_t = 2$ task-level summaries. Furthermore, memory updates (e.g., insertion, modification, or deletion) are applied only to the top $N_{\text{update}} = 10$ most relevant entries in the Semantic Memory to ensure efficiency and coherence.

## G Supplementary Examples for Qualitative Analysis

### G.1 Real World

In Figure 9, we demonstrate an example of RoboMemory learning through trial and error in a real-world environment. Our task is "place a banana into the oven." This task required RoboMemory to complete the objectives of finding the banana, picking it up, and transporting it to the oven. We observed that RoboMemory became stuck in an infinite loop during the first attempt. The banana was randomly placed on the "kitchen counter," but RoboMemory overlooked this navigation target and remained trapped, exploring other navigation targets instead.

However, based on this bad attempt, the semantic memory summarized that the robot should not repeatedly search in locations where the "banana" could not be found. Meanwhile, the episodic memory recorded what RoboMemory had done and the outcomes during the first attempt. Based on the information provided by semantic and episodic memory, in the second attempt, RoboMemory recognized that it had not previously tried navigating to the "kitchen counter." After attempting this, it successfully completed the task. This example illustrates the role of RoboMemory's long-term memory.

We also provide an example that completes the task in the first attempt. The example is shown in Figure 10. This example demonstrates that the RoboMemory has the ability to handle some relatively complex tasks in the real world. The task in this example is "Place a box of gum into the basket and put the basket on the kitchen counter". Because two objects in different positions are involved in this task, RoboMemory has to memorize the position of at least one object to achieve the goal. With the help of the spatial memory, RoboMemory completes the task successfully.

**Task: Place banana into the oven**

**The first time**

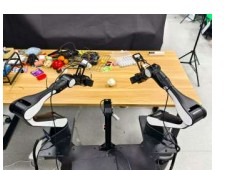 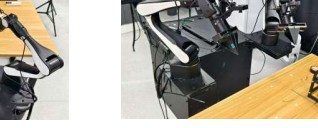 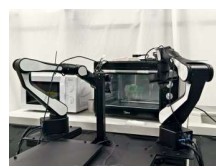

**Step 1**: Navigate to the left side of the desk

**Step 2**: Navigate to the right side of the desk

**Step 3**: Navigate in front of the oven

**Step 4~15**: Navigate to these three points in an infinite loop, but failed to explore new areas.

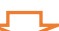

**Semantic Memory** … The robot should avoid navigating to places that do not contain the target object again and again.

**Episodic Memory** … The task is to place a banana into the oven. The robot navigated to the left side of the desk, the right side of the desk, and in front of the oven multiple times, but failed to find the banana.

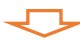

**The second time**

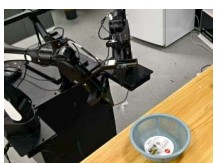 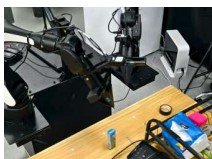 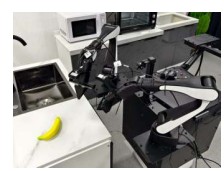 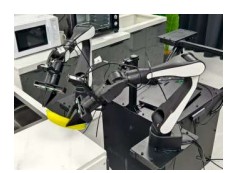

**Step 1**: Navigate to the left side of the desk

**Step 2**: Navigate to the right side of the desk

**Step 3**: Navigate to the kitchen counter

Step 4: Pick up the banana

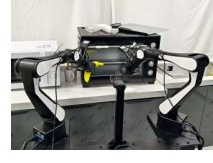 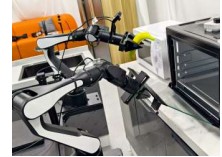 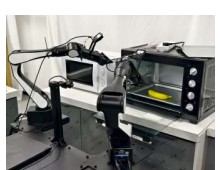 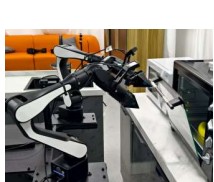

**Step 5**: Navigate to the oven

**Step 6**: Open the oven

**Step 7**: Put down (banana) to the oven

**Step 8**: Close the oven

Figure 9: Case that a task is failed, but the experience can help RoboMemory to succeed in the next try.

## G.2 EB-ALFRED

We select three examples in EB-ALFRED to show the errors that RoboMemory may encounter and the reasons why or why not RoboMemory can achieve the goal.

### G.2.1 SUCCESSFUL EXAMPLE

We select a successful example to show how RoboMemory performed in the EB-ALFRED environment. The example trajectory is shown in Figure 11.

**Task: Place a box of gum into the basket and put the basket to the kitchen counter**

**Step 1**: Navigate to the left side of the desk

**Step 2**: Navigate to the kitchen counter

**Step 3**: Pick up the gum box

**Step 4**: Navigate to the left side of the desk

**Step 5**: Put down (gum box) to the basket

**Step 6**: Pick up the basket

**Step 7**: Navigate to the kitchen counter

**Step 8**: Put down (basket) to the kitchen counter

Figure 10: Case that a task is successful.

The task of this example is "set a plate with a spoon on it on the kitchen table". However, in step 10, the Planner seems to ignore the temporal information from memory modules. RoboMemory thinks that it still needs to pick up the spoon (even though it has already placed a spoon in the plate). However, with the help of the critic, it finally becomes aware that picking up another spoon is redundant, so RoboMemory goes back to the current trajectory and successfully completes the task at the end.

In this example, RoboMemory successfully overcame the hallucination and eventually achieved the goal. This example demonstrates that the critic module can help RoboMemory to overcome error cases.

### G.2.2 FAILED EXAMPLE

We demonstrate a representative example of the Critical Error. The example trajectory is shown in Figure 12. In this example, the task involves slicing and heating a tomato and moving the heated tomato slice to the trash can. Initially, RoboMemory successfully sliced the tomato with a knife. But when the planner plans the whole sequence, it forgets to drop the knife before picking up the tomato (this is necessary because in EB-ALFRED, the robot can only hold one object at a time). The critic and the planner should notice this situation and ask the critic to replan, as RoboMemory failed to pick up a tomato slice. However, the critic module ignores this issue, and thus, after it heats the knife instead of a tomato slice, it stacks in an infinite loop.

Besides, we provide another example demonstrating a representative failure caused by inaccurate action planning. The example trajectory is shown in Figure 13. In the trajectory, RoboMemory is asked to place two CDs into the drawer. However, at step 6, the robot failed to select correct CD object. In this experiment, RoboMemory has already put CD_2 into the drawer, but it keeps picking up CD_2 even though the memory has clearly indicated that CD_2 has already been put down. So we classify this as inaccurate action error. This indicates that the planner failed to comprehensively integrate information from both the memory and information-gathering modules, resulting in inaccurate action planning.

**Task: Set plate with a spoon in it on the kitchen table**

Figure 11: Case that a task is successful with the help of the critic and spatial memory modules.

## Task: Cook a sliced tomato and throw it in the trash

**Step 1**: Initialize

**Step 2**: Find the tomato

**Step 3**: Find the knife

**Step 4**: Pick up the knife

**Step 5**: Slice the tomato

**Step 6**: Find tomato

**Step 7**: Slice the tomato

**Step 8**: Pick up the tomato

**Step 9**: Find the microwave

**Step 10**: Open the microwave

**Step 12**: Put down

**Step 13 ~ 30**: Infinite loop

Figure 12: Case that a task fails in an infinite loop because the critic module failed to stop the agent when its planned action is no longer suitable.

**Task: Move two CDs to the bottom drawer of the desk**

**Step 1**: Find CD_2

**Step 2**: pick up CD_2

**Step 3**: Find the drawer

**Step 4**: Open the drawer

**Step 5**: Put down

**Step 6**: Find the CD_2

**Step 7**: Pick up CD_2

**Step 8**: Put down

**Step 9**: Find CD_2

**Step 10**: Pick up CD_2

· · ·

**Step 11~30**: Infinite loop

Figure 13: Case that a task fails in an infinite loop because of inaccurate action planning.

