# OpenReview forum: "RoboMemory: A Brain-inspired Multi-memory Agentic Framework for Interactive Environmental Learning in Physical Embodied Systems"
_ICLR.cc/2026/Conference — Submitted to ICLR 2026_

### Official Review · Reviewer_tMhd · 2025-10-29

**Soundness:** 3
**Presentation:** 2
**Contribution:** 3
**Rating:** 4
**Confidence:** 3

**Summary:**

This paper presents RoboMemory, a novel framework designed to equip embodied agents with a sophisticated, brain-inspired memory system. The core contribution is a unified architecture that integrates Spatial, Temporal, Episodic, and Semantic memory modules, operating in parallel to enable long-horizon planning and interactive learning in dynamic, partially observable environments. The results demonstrate a significant performance improvement over strong baselines, on both simulated benchmarks and a real-world robotic environment.

**Strengths:**

- Novel and Well-Motivated Architecture: The brain-inspired design, mapping functional components (hippocampus, prefrontal cortex, etc.) to agent modules, is a compelling and timely approach. The parallelized, multi-memory system addresses a clear gap in the literature, moving beyond simple context buffers or single-memory-type frameworks.
- Benchmark Performance: Demonstrating a solid improvement over the base methods on both EB-ALFRED and EB-Habitat benchmarks.
- Real-World Deployment: The real-world experiments are a significant strength. Showing performance improvement between the first and second attempt on the same tasks without memory reset is a powerful, practical demonstration of interactive environmental learning.

**Weaknesses:**

- The paper is written on a relatively high-level, without enough technical details. For example, the contents of Figure 1 and Figure 2 is somewhat overlapped, while there is not enough detailed formula, or framework figure to introduce the methodology. Especially, in Secton 2.2.1 and 2.2.2, it is hard to understand how the four memories (and the KG) are computed and updated, how they are rertrieved, and how they can guide the base model to yield the outputs. Furthermore, a detailed example (with completed set of query, action, four memories) may help the reader understand what hapepend.
- The paper's organization needs to be improved. Modules which have been proposed in previous works (e.g. RAG, planner, executor) are suggested to be included in the Preliminary, while the core contribution (the memory modules) can exhibit more method details; Line 253-270 is confusing. Is it better to combine 2.2.1 and 2.2.2 together, while having a separate section about the memory insert&update mechanism?
- Besides inferencing the VLM models, other baselines include Voyager, Reflexion, and Cradle. However, Voyager is out-of-date (2023) while Reflexion is primarily to solve the general NLP tasks, such as MBPP and Humaneval. The baselines need to be strengthened by more recent and robotic-focused methods.

**Questions:**

- Why **Temproral and Spatial Memory** and **Episodic and Semantic Memory** are seperated? A summarized, abstracted memory sample   can include all the above features. Without detailed information, it is hard to ensure that all these memory types are necessary.
- How the memories, the KG, the retriever and the base model (Qwen-VL) interact with each other? there should be more details to make the paper self-contained.
- On Line 279, it is said that the first step is not evalulated by the Critic. However, the first step of planning seems to be followed by a critic in Figure 2.
- Efficiency analysis compares different detailed implementations of RoboMemory. How about its efficiency compared to previous baselines? Is there any theoretical observations on the efficiency?

---

> ### Author Response · Authors · 2025-11-20
>
> We thank the reviewer for identifying areas where our writing and structure could be improved. We have extensively revised the paper based on your advice.
>
> ----
>
> > W1 & W2: Unclear writing, missing algorithm details, and interaction logic between different modules is unclear.
>
> **A1 & A2:** We appreciate this feedback and have made significant revisions.
>
> - First of all, we have completely rewritten Section 3 (Section 2 for the original manuscript) to include detailed descriptions of the update and retrieval algorithms for all four memory modules, clarifying design choices.
>
> - Secondly, we use a separate section to describe the update and retrieval process of memory systems in summary.
>
> - Moreover, we added **Appendix B**, which contains pseudocode providing a comprehensive step-by-step view of how the RoboMemory modules collaborate with each other to complete a task. See Algorithm 1 for more details.
>
> ----
>
> > W3: Baselines are outdated.
>
> **A3:** We have added **RoboOS [1]** as a new baseline and included its results in the main comparison table (Table 1). The table is shown below:
>
> | Method | Type | Average SR | Average GC | EB-ALFRED Base SR | EB-ALFRED Base GC | EB-ALFRED Long SR | EB-ALFRED Long GC | EB-Habitat Base SR | EB-Habitat Base GC | EB-Habitat Long SR | EB-Habitat Long GC |
> |--------|------|------------|------------|-------------------|-------------------|-------------------|-------------------|--------------------|--------------------|--------------------|--------------------|
> | RoboOS (Qwen2.5-VL-72B-Ins) | Baselines | 25.5 | 33.0 | 32.0 | 38.4 | 12.0 | 17.6 | 38.0 | 47.8 | 20.0 | 28.2 |
> | RoboOS (RoboBrain2-32B) | Baselines | 20.0 | 25.7 | 32.0 | 37.2 | 8.0 | 13.2 | 28.0 | 34.8 | 12.0 | 17.4 |
> | **RoboMemory (Qwen2.5-VL-72B-Ins)** | **Ours** | **70.5** | **79.7** | 68.0 | **75.5** | 66.0 | **81.3** | 86.0 | 88.0 | 62.0 | **74.0** |
>
>
> ----
>
> > Q1: Why distinguish between Spatial-Temporal and Episodic-Semantic memory?
>
> **A1:** Different information requires different data structures. Approaches like Reflexion or Cradle try to summarize everything into a single abstract text entry. LLMs struggle to accurately summarize complex spatial relationships into text, and equally struggle to reconstruct spatial layouts from such text. To address this problem, a dedicated spatial module (using KGs) addresses the gap in spatial understanding that text-based memory lacks. Meanwhile, for Semantic and Episodic Memory, using a different data structure (Vector DB) is more suitable for long-term memory. To sum up, because different memory types require different data structures and the VLM planner may fail to extract various types of information within a single unified memory module, we need to use different memory modules to store different types of data.
>
> ---
>
> > Q2: How do the memories, KG, retriever, and base model interact?
>
> **A2:** We have visualized this interaction in **Figure 2**. Additionally, as mentioned above, we have included pseudocode in **Appendix B** to provide a technical detail of this coordination.
>
> ----
>
> > Q3: Line 279 says the first step is not evaluated by Critic, but Figure 2 implies otherwise.
>
> **A3:** Thank you for spotting this inconsistency. We have corrected **Figure 2**. In the figure, **Green** indicates the action is executed successfully. **Red** indicates the Critic blocking the execution of the action and triggering a re-plan. **Blue** indicates a re-planned action. The diagram now accurately reflects the logic of the Planner-Critic mechanism.
>
> ----
>
>
> > Q4: Is there any theoretical analysis on efficiency?
>
> **A4:** Yes, we have added a theoretical analysis in **Appendix D.3**. We prove that the Knowledge Graph update algorithm has an upper bound of $O(D^K)$, where $D$ is the maximum node degree and $K$ is the hop count (usually small, e.g., 2 or 3). Due to the sparsity of spatial graphs, this is highly efficient. Besides, Semantic and Temporal memory updates only involve a bounded number of items (constant complexity). This theoretical foundation guarantees the efficiency of our memory updates. Comparison with baselines regarding efficiency will be detailed in future work.
>
> ----
> We hope our responses have addressed your concerns.
> ----
>
> [1] Tan, H., Hao, X., Chi, C., Lin, M., Lyu, Y., Cao, M., ... & Zhang, S. (2025). Roboos: A hierarchical embodied framework for cross-embodiment and multi-agent collaboration. arXiv preprint arXiv:2505.03673.

---

### Official Review · Reviewer_DhZZ · 2025-10-31

**Soundness:** 2
**Presentation:** 3
**Contribution:** 2
**Rating:** 6
**Confidence:** 3

**Summary:**

The authors present RoboMemory, a framework inspired by human brain to unify spatial, temporal, episodic and semantic memory for long-horizon planning and interactive learning.

RoboMemory has:
- an information processor for generating textual summary of the current scene using a Step Summarizer and a Query generator;
- a memory system with object relations and layout in spatial-temporal memory and action histories and experiences in a long-term memory. An episodic and semantic memory system is also used for interactive learning. All these use a RAG extractor, updater and storage.
- a closed-loop planning module which uses retrieved memories for high-level actions, which uses Planner-Critic modules;
- a low-level VLA executor fine-tuned with LoRA and a SLAM-based navigation system.

The spatial information is represented as Spatial Knowledge Graphs in the memory, which is updated as new observations come in.

They experiment with EmbodiedBench-ALFRED and EB-Habitat benchmarks and show that an open-source Qwen2.5-VL-72B with RoboMemory performs better than closed-source SOTAs, and several other baselines from GPT, Claude, InternVL families, etc. They also compare against VLM-Agent frameworks like Voyager, Reflexion, and Cradle.  The authors also perform real-world experiments to showcase the capacity of RoboMemory for interactive learning as it performs better the second-time it is deployed in the environment. They also conduct ablation studies to study value of different aspects of RoboMemory. The authors find that the spatial memory is the most important, followed by the critic and long-term memory.

**Strengths:**

1. The design of the approach is very careful and well-inspired. The idea of borrowing knowledge from human-brain working is interesting, and the approach shows how to model this using MLLMs.
2. The authors test against various open-source and closed-source VLMs, as well as SOTA VLM-Agent framework ensuring a comprehensive evaluation.
3. The authors also show that their approach is effective in interactive learning in teh real-world which is a useful experiment.

**Weaknesses:**

1. While the design is careful, it may be possible to simplify the approach by combining various aspects of the different memories created.
2. The authors do not discuss the hardware, costs and compute required for their approach against the baselines methods. This method may achieve success, but because of the several VLMs and VLAs involved, it may consume a lot of resources.
3. The real-world evaluation is done at a smaller-scale, with only 15 tasks and the success rates are still pretty-low (46.67%). The authors do not provide any baselines for the real-world tasks.

**Questions:**

1. Is the VLA-executor trained for the real-world tasks? If yes, how?
2. What are some ways the issues with VLMs and VLAs in the real-world can be addressed?

---

> ### Author Response · Authors · 2025-11-20
>
> We thank the reviewer for recognizing the value of our work. Here are our responses to your queries.
>
> ----
>
> > W1: The Memory System could be simplified by integrating different memory modules.
>
> **A1:** We appreciate this suggestion. We have explored different approaches to merging memory modules for simplification. But we found out that our design is necessary because of the following reasons:
>
> - **1. Experimental Failure:** We attempted to merge different memory modules, such as Semantic and Episodic Memory, but observed a significant decline in performance. This may be because current VLMs can not update memories with a mix of different memory types.
>
> - **2. Efficiency:** Since memory updates occur in parallel, decreasing the number of memory modules does not significantly improve efficiency (verified in Section 4.4).
>
> - **3. Cognitive Basis:** Our design is based on classifications in cognitive neuroscience, which justifies the use of distinct memory types.
>
> ----
>
> > W2: More discussion on cost and resources.
>
> **A2:** We acknowledge that although our parallelization design can increase the efficiency of the Memory system. However, computational cost is not benefited from a parallelization design. So computational cost is not an advantage of RoboMemory, but we believe it is fair.
>
> - First of all, RoboMemory can achieve or exceed closed-source model performance using open-source models. This significantly reduces API costs and enables local deployment.
>
> - Even if current costs are high, the architectural validation is valuable. We are actively working on distilling these modules into smaller models to reduce resource usage in the future.
>
> ----
>
> > W3.1: Real-world evaluation scale is small and success rate is low.
>
> **A3.1:** We acknowledge the limitations of the real-world experiments. However, testing long-horizon tasks containing both navigation and manipulation on mobile robots is challenging. So the test size may be relatively small. However, we do lots of experiments in a virtual benchmark (200 tasks) instead.
>
> The primary reason for the high success rate of RoboMemory in real-world scenarios is due to two key factors. First of all, the failure of the low-level executer. Even with successful planning, the VLA-based low-level executor may fail to execute. Furthermore, the Mobile-ALOHA fixed camera angle limits the sensing ability of RoboMemory. RoboMemory may fail to be aware of some important information because of the limited camera field of view.
>
> We want to emphasize that despite these physical limitations, the experiments successfully validated the system's interactive learning capability in a real environment.
>
> ----
>
> > W3.2: No comparison with Baseline in the real environment.
>
> **A3.2:** We omitted this because most existing baselines are not designed for real-world robots. They may not be adapted to our robot hardware. This problem may make the comparison unfair. In contrast, the virtual environment provides a standardized arena for comparison. Besides, we add  **RoboOS [1]**, which is originally tested on real-world settings, as our baseline in real-world environments.

---

> ### Author Response · Authors · 2025-11-20
>
> > Q1: Was the VLA trained?
>
> **A1:** Yes, we trained the VLA to adapt to our robot. The Full training details are in **Appendix E.2**. In summary, we utilized an AdamW optimizer, with a batch size of 32x6, and trained for 10,000 steps on a dataset comprising 1,040 episodes across various tasks (e.g., faucet operation, picking/placing objects). The detail settings are shown below:
>
> | Action Type | #Episodes | Parameter | Value |
> | :--- | ---: | :--- | :--- |
> | Turning on/off faucet | 142 | Optimizer | AdamW |
> | Picking up & Placing basket on counter | 63 | Batch size | 32×6 |
> | Picking up & Placing basket in sink | 72 | Training steps | 10,000 |
> | Picking up & Placing banana into basket | 114 | Learning rate | 6.12×10−5 |
> | Throwing bottle into trash bin | 132 | warm up step | 500 |
> | Placing gum box on dish | 120 | LoRA Configuration | |
> | Picking up & Placing cup on plate | 51 | rank | 16 |
> | Picking up & Placing dish into sink | 69 | α | 16 |
> | Throwing paper ball into trash bin | 135 | Resource Usage | |
> | Open/close oven | 142 | GPU | A100-80GB × 6 |
> | Total episodes | 1040 | Training time | 12 hours |
>
> ----
>
> > Q2: What are some ways the issues with VLMs and VLAs in the real world can be addressed?
>
> **A2:** As shown in w3.1. and Section 5 (Conclusion and future work). Their are several problems of current VLA and VLM:
>
> - 1. The camera is fixed, so some important information may be ignored by RoboMemory because it can not be seen.
>
> - 2. The limitation of the communication method between VLM and VLA. Currently, the instruction pass from VLM to VLA is based on language. But some instructions may be hard to express in language or text-based function calls. So, exploring a better API for VLM/VLA communication is still needed.
>
> -----
> We hope our responses have addressed your concerns.
> ----
>
> [1] Tan, H., Hao, X., Chi, C., Lin, M., Lyu, Y., Cao, M., ... & Zhang, S. (2025). Roboos: A hierarchical embodied framework for cross-embodiment and multi-agent collaboration. arXiv preprint arXiv:2505.03673.

---

> ### Author Response · Authors · 2025-12-03
>
> > Thanks to reviewer DhZZ for pointing out the issue of lacking a Baseline for real-world settings, we added RoboOS as our real-world experiment baseline. The result is shown in Figure 5 of the updated version. For your convenience, we provide our results below.
>
> | RoboOS (Qwen) | RoboOS (RoboBrain) | RoboMemory (First attempt) | RoboMemory (Second attempt) |
> | :---: | :---: | :---: | :---: |
> | 20.00% | 6.67% | 26.67% | **46.67%** |

---

### Official Review · Reviewer_Ni9E · 2025-11-01

**Soundness:** 2
**Presentation:** 3
**Contribution:** 2
**Rating:** 2
**Confidence:** 4

**Summary:**

A method called RoboMemory is proposed for creating an agent with memory based on a visual-language model for embodied artificial intelligence tasks.

**Strengths:**

1. The paper is well-written and easy to read.
2. It explores an important direction in developing embodied agents with memory mechanisms.
3. Experiments are conducted on a real robot.
4. The method demonstrates improvements on the tasks considered.

**Weaknesses:**

1. The **Related Works** section (at least in a shortened version) should be included in the main text. Its purpose is to position the work relative to existing approaches and highlight its novelty and relevance.
2. There is no comparison of the proposed method with other approaches on a real robot.
3. Experiments on tasks that truly require memory, beyond spatial memory, are missing. The authors should at least propose a small set of test tasks and demonstrate comparisons on them with a detailed analysis of the results. As an example, benchmarks proposed in the RL/VLA field for memory-intensive robotic tasks [1, 2] could be used.
4. No baseline using spatial memory is employed, even though the main improvement in the proposed model comes from it. A baseline with spatial memory should be added, for example, RoboOS [3].

I am willing to revise my assessment if the mentioned shortcomings are addressed.

**References:**
1. Fang, Haoquan, et al. "Sam2act: Integrating visual foundation model with a memory architecture for robotic manipulation." arXiv preprint arXiv:2501.18564 (2025).
2. Cherepanov, Egor, et al. "Memory, Benchmark & Robots: A Benchmark for Solving Complex Tasks with Reinforcement Learning." arXiv preprint arXiv:2502.10550 (2025).
3. Tan, Huajie, et al. "Roboos: A hierarchical embodied framework for cross-embodiment and multi-agent collaboration." arXiv preprint arXiv:2505.03673 (2025).

**Questions:**

1. The tasks in EB-ALFRED generally do not require memory, except for spatial memory, which involves knowing the location of objects (as also confirmed by the results in Table 2). How appropriate is it to evaluate the proposed framework, where memory mechanisms are the main contribution, on such tasks?
2. How does the proposed approach compare to methods that use scene representation as a graph (which can be considered a form of spatial memory) [1, 2, 3, 4]? They should be included as baselines.

**References:**
1. Honerkamp, Daniel, et al. "Language-grounded dynamic scene graphs for interactive object search with mobile manipulation." IEEE Robotics and Automation Letters (2024).
2. Ekpo, Daniel, et al. "Verigraph: Scene graphs for execution verifiable robot planning." arXiv preprint arXiv:2411.10446 (2024).
3. Onishchenko, Anatoly, Alexey Kovalev, and Aleksandr Panov. "LookPlanGraph: Embodied instruction following method with VLM graph augmentation." Workshop on Reasoning and Planning for Large Language Models.
4. Tang, Yujie, et al. "Openin: Open-vocabulary instance-oriented navigation in dynamic domestic environments." IEEE Robotics and Automation Letters 10.9 (2025): 9256-9263.

---

> ### Author Response · Authors · 2025-11-20
>
> We are grateful to Reviewer Ni9E for the constructive feedback. These comments are helpful in improving our paper. We have incorporated your suggestions into the revised version and provided a point-by-point response to your questions. We hope that our responses will address your concerns.
>
> ---
>
> > W1: Related work is in the Appendix.
>
> **A1:** Thank you for the suggestion. We have relocated the **Related Work** section to the main body of the paper. Please see **Section 2**.
>
> ---
>
> > W2: Comparison with Baselines on real robots.
>
> **A2:** We mainly focused on virtual comparison for baselines for the following reasons:
>
> - 1. Firstly, the comparison in the virtual environment effectively demonstrates the advantages of RoboMemory over other settings.
>
> - 2. Secondly, Most existing baselines are either not deployed on real hardware or are technically difficult to adapt to our specific Low-level Executer.
>
> - 3. To address your concern, we add RoboOS [1], which was originally tested on real-world settings, as our baseline in real-world environments.
>
> --------------
>
> > W3: The benchmark used lacks demand for memory types beyond spatial memory.
>
> **A3:** We respectfully argue that the chosen benchmarks and our experiments validate the need for multiple memory types.
>
> - First of all, we believe that the EB-ALFRED benchmark can show the importance of our memory systems. In our ablation study, removing each memory item results in a drop in performance.
>
> - Secondly, current VLA/RL systems (e.g., Pi0) can not accept complex, abstract Memory conditions in text form as input. We need to do further training to enable VLAs to collaborate with our memory systems. Therefore, they are not currently suitable for this specific Memory framework.
>
> - However, enabling VLAs to utilize such memory is a key direction for our future work.
>
> -------------
>
> > W4: Supplement spatial memory baselines.
>
> **A4:** We have added **RoboOS** (tested with both RoboBrain2.0-32b and Qwen2.5-72b) as a spatial baseline. Our tests show that RoboOS can not beat RoboMemory in performance due to the lack of a Planner-Critic module, the absence of long-term memory, and most importantly, the simple scene-graph base spatial memory is not able to handle diverse spatial relationships. This experiment highlights the advantages of Planner-Critic and KG-based Spatial Memory design in RoboMemory. The result is shown below.
>
> ------------------
>
> > Q1: EB-ALFRED Benchmark has low demand for memory mechanisms, especially non-spatial ones.
>
> **A1:** As we answered in A3, we believe our testing in virtual benchmarks places a high demand on comprehensive memory for several reasons:
>
> - **1. Task Complexity:** According to EmbodiedBench [2], tasks will involve long-term manipulation of multiple objects in a **partially observable environment** in different positions. To meet these demands, EB-ALFRED requires beyond spatial memory. For example, **Temporal Memory** is responsible for recording the interaction history of RoboMemory. The historical information enables RoboMemory to determine which subgoal has already been achieved and which subgoal still needs to be completed. Besides, our ablation study (Table 2) confirms that all memory modules contribute to performance.
>
> - **2. Adding EB-Habitat:** In the revised version, we have moved the performance comparison between baseline methods and RoboMemory in EB-Habitat benchmarks to the main text (Table 1). This benchmark involves high-level navigation and object finding, which require powerful memory systems.
>
> ------
>
> > Q2: The difference between RoboMemory's mechanism and traditional Scene-graph mechanisms.
>
> **A2:** As discussed in A4, there are distinct differences between our KG-based spatial memory and scene-graph-based spatial memory:
>
> - **1. Structure & Diversity:** Traditional scene graphs (like those in RoboOS) are often tree-structured, compared to a Knowledge graph. A Tree can only store a limited type of relationships. However, our approach can record different spatial relationships between objects if they can be described in language.
>
> - **2. VLM-Generated KG:** Our approach uses a VLM to generate a Knowledge Graph. This allows the system to capture and master a much more diverse set of spatial relationships, the effectiveness of which is proven by our comparison with RoboOS.
>
> - **3. Dynamic update:** One big advantage of KG-based spatial memory in RoboMemory is that the KG can be generated from scratch. As the robot explores the environment, the KG can dynamically update and expand. See Figure 8. In Section E.2.
>
> ----
> We hope our responses have addressed your concerns.
> ----

---

> ### Author Response · Authors · 2025-11-20
>
> | Method | Type | Average SR | Average GC | EB-ALFRED Base SR | EB-ALFRED Base GC | EB-ALFRED Long SR | EB-ALFRED Long GC | EB-Habitat Base SR | EB-Habitat Base GC | EB-Habitat Long SR | EB-Habitat Long GC |
> |--------|------|------------|------------|-------------------|-------------------|-------------------|-------------------|--------------------|--------------------|--------------------|--------------------|
> | RoboOS (Qwen2.5-VL-72B-Ins) | Baselines | 25.5 | 33.0 | 32.0 | 38.4 | 12.0 | 17.6 | 38.0 | 47.8 | 20.0 | 28.2 |
> | RoboOS (RoboBrain2-32B) | Baselines | 20.0 | 25.7 | 32.0 | 37.2 | 8.0 | 13.2 | 28.0 | 34.8 | 12.0 | 17.4 |
> | **RoboMemory (Qwen2.5-VL-72B-Ins)** | **Ours** | **70.5** | **79.7** | 68.0 | **75.5** | 66.0 | **81.3** | 86.0 | 88.0 | 62.0 | **74.0** |
>
> ----------------
>
>
> [1] Tan, H., Hao, X., Chi, C., Lin, M., Lyu, Y., Cao, M., ... & Zhang, S. (2025). Roboos: A hierarchical embodied framework for cross-embodiment and multi-agent collaboration. arXiv preprint arXiv:2505.03673.
>
> [2] Yang, R., Chen, H., Zhang, J., Zhao, M., Qian, C., Wang, K., ... & Zhang, T. (2025). Embodiedbench: Comprehensive benchmarking multi-modal large language models for vision-driven embodied agents. arXiv preprint arXiv:2502.09560.

---

> ### Author Response · Authors · 2025-11-27
>
> Dear Reviewer Ni9E,
>
> We sincerely appreciate the time and effort you invested throughout the review process and fully understand the demands on your schedule. We have included additional details, including testing RoboOS as a baseline, in the hope that they address your concerns.
>
> As the rebuttal deadline approaches, we would be grateful to know if you have any further questions or concerns. We will do our best to provide any additional clarification.
>
> Best regards,
> Authors of the Paper 19655

---

> ### Author Response · Authors · 2025-12-03
> **Additional Real-world Baseline added.**
>
> > Thanks for reviewer Ni9E to points out W2. To address W2, we add RoboOS as our real-world experiment baseline. The result is shown in Figure 5 of the updated version. For your convinence, we provide our result below.
>
> | RoboOS (Qwen) | RoboOS (RoboBrain) | RoboMemory (First attempt) | RoboMemory (Second attempt) |
> | :---: | :---: | :---: | :---: |
> | 20.00% | 6.67% | 26.67% | **46.67%** |

---

### Official Review · Reviewer_ct4h · 2025-11-02

**Soundness:** 1
**Presentation:** 2
**Contribution:** 2
**Rating:** 4
**Confidence:** 3

**Summary:**

RoboMemory proposes a brain-inspired embodied agent architecture integrating multiple memory systems under a unified framework. It aims to bridge high-level planning (VLM-based) and low-level execution (VLA-based) for long-horizon and real-world tasks. The architecture has several computational modules, to do closed-loop planning with dynamic memory updating. Experiments show improvements over open-source VLM baselines (e.g., +25% over the Qwen2.5-VL-72B base) on EmbodiedBench and real-world robotic tasks.

**Strengths:**

* The paper tackles an important problem: improving long horizon reasoning in complex embodied tasks.

* The paper reports results in multiple environments, including the real world

**Weaknesses:**

1) I have several doubts about the experimental methodology in this paper. If I understand correctly, all the VLM baselines in the paper (or atleast the open-source ones) have zero historical awareness since they only take one frame in, but it is trivial and necessary to compare to a fairer baseline which takes multiple frames across the history as input to give it some required context - since that is something current day VLMs support easily. Is my understanding correct that this is not currently done for the baselines? If not, what is supposed to be the main takeaway of the paper? Since any method would struggle if only provided a single visual frame for decision making.

2) Why isn’t a reasoning model or a strong VLM (like GLM 4.5V thinking) used as part of the baselines (with the above adjustment of using multiple input frames).

3) There seem to be no reporting of confidence intervals or results averaged across seeds. The authors should ensure such results are reported before we can actually interpret the results with some confidence.

4) There is a lot of scaffolding proposed in the paper which introduces a lot of overhead (and isn’t inherently novel, there are many many papers doing similar scaffolds for visual tasks like question answering based on videos for example) - and makes the contribution of the paper hard to understand.

5) Are the results which are presented on the EB-Alfred task in the paper taken from the EmbodiedBench paper directly? Why don’t they match the numbers there?

6) Why do the Habitat results not contain the other baselines used in the EmbodiedBench paper and also in this paper on the Alfred task?

**Questions:**

Listed above.

---

> ### Author Response · Authors · 2025-11-20
>
> We sincerely thank the reviewer for their insightful questions regarding our paper. These comments have significantly helped us improve the quality of our paper. We have revised the manuscript accordingly and hope our responses address your concerns.
>
> ---------------
>
> > W1: The baseline lacks memory entirely, which may lead to an unfair comparison.
>
> **A1:** We appreciate the reviewer bringing this to our attention. We have clarified this in the revised manuscript (Appendix F.2). Every baseline includes memory systems capable of storing interaction history. For the Single VLM Agent, we followed the official settings outlined in the **EmbodiedBench paper [1]**. The baseline utilizes a text-based interaction history. For other baselines, all other selected baselines include memory systems. Especially, for the **Cradle [2]** baseline, we adopted their original setting, which uses three frames as context, providing it with both visual context and memory capabilities. As every baseline includes a memory mechanism, we believe that our comparison is fair and valid. The results demonstrate the specific advantages of our memory system against these baseline methods.
>
> -------------------------
>
> > W2: Reasoning models were not included in the comparison.
>
> **A2:** Thank you for this suggestion.
>
> - First of all, the original settings of EmbodiedBench [1] were designed for non-reasoning models. To ensure fairness, we initially selected non-reasoning backbones.
>
> - Meanwhile To address your concern, we have added **RoboOS** (powered by **RoboBrain2.0-32b [3]**) as a new baseline. RoboBrain2.0-32b is an RL-finetuned model. Our results show that thanks to RL fine-tuning, it approaches the performance level of the larger Qwen2.5-72B-Instruct. The result is shown below.
>
> - Lastly, we plan to conduct further testing with reasoning models using EmbodiedBench settings in our future work.
>
> ---------------------------
>
> > W3: Confidence intervals were not reported.
>
> **A3:** We apologize for not clarifying this in the original text. We utilized the same statistical method as EmbodiedBench. Because we set the model **Temperature = 0** for deterministic evaluation, we do not think it necessary to report confidence intervals. To clarify this, we have added a description of this setting in **Section 4.2** of the updated manuscript.
>
>
> -------------------------------
>
> > W4: Concerns regarding the novelty of the model framework and efficiency.
>
> **A4:** We appreciate this opportunity to clarify our contributions.
>
> - First of all, one of our core innovations is the parallel architecture for memory frameworks. As shown in **Figure 3**, while the Memory update includes complex operations (like Spatial KG updates), parallelization ensures that the overall system latency increases only marginally compared to simple Temporal Memory updates.
>
> - Secondly, the designs of the Spatial-Temporal Memory and the dynamic Spatial KG update algorithm are designed for the Embodied Agent. While we acknowledge that the absolute time cost is currently high, the architectural innovation of deploying a comprehensive memory system in an embodied agent is significant.
>
> - Besides, we plan to replace individual modules with smaller, post-trained models in future work to reduce absolute latency.
>
>
> ----------------------------------------
>
> > W5: Are the results which are presented on the EB-Alfred task in the paper taken from the EmbodiedBench paper directly? Why don’t they match the numbers there?
>
> **A5:** Thank you for pointing this issue out. We only select a subset of the EB-ALFRED benchmark. The reason why we only select a subset of EB-ALFRED is because:
>
> -  We choose subset of EB-ALFRED that focus on testing agent's planning ability. As noted in Section 4.1 (lines 372-375), we selected the **Base** and **Long** subsets of EB-ALFRED and EB-Habitat to test RoboMemory because they are specifically designed to test the agent’s **planning ability**.
>
> - Besides, other subsets of EB-ALFRED and EB-Habitat, such as the "Common" set in EB-ALFRED, are conceptually equivalent to the **Base** set. Still, the task description is more challenging (e.g., the task description in the "Common" set: "Refer to objects indirectly using common sense knowledge" [1]). These test sets test the model's semantic understanding rather than the Agent Memory framework itself. Therefore, using the selected subsets is a strategic trade-off that allows us to focus on our core research objectives.
>
> ------------------------------------
>
> > W6: Missing Habitat experimental results and Baselines.
>
> **A6:** We apologize for the confusion caused by the omission. We originally omitted the full Habitat results due to space constraints. Taking advantage of the extra page allowed during the rebuttal, we have added the complete comparison from the appendix to the main text and combined the success rates with EB-ALFRED in **Table 1**.
>
> ---
> We sincerely hope your concerns are fully addressed.
> ---

---

> ### Author Response · Authors · 2025-11-20
>
> **Result of RoboOS:**
>
> | Method | Type | Average SR | Average GC | EB-ALFRED Base SR | EB-ALFRED Base GC | EB-ALFRED Long SR | EB-ALFRED Long GC | EB-Habitat Base SR | EB-Habitat Base GC | EB-Habitat Long SR | EB-Habitat Long GC |
> |--------|------|------------|------------|-------------------|-------------------|-------------------|-------------------|--------------------|--------------------|--------------------|--------------------|
> | RoboOS (Qwen2.5-VL-72B-Ins) | Baselines | 25.5 | 33.0 | 32.0 | 38.4 | 12.0 | 17.6 | 38.0 | 47.8 | 20.0 | 28.2 |
> | RoboOS (RoboBrain2-32B) | Baselines | 20.0 | 25.7 | 32.0 | 37.2 | 8.0 | 13.2 | 28.0 | 34.8 | 12.0 | 17.4 |
> | **RoboMemory (Qwen2.5-VL-72B-Ins)** | **Ours** | **70.5** | **79.7** | 68.0 | **75.5** | 66.0 | **81.3** | 86.0 | 88.0 | 62.0 | **74.0** |
>
> -------------------
>
> [1] Yang, R., Chen, H., Zhang, J., Zhao, M., Qian, C., Wang, K., ... & Zhang, T. (2025). Embodiedbench: Comprehensive benchmarking multi-modal large language models for vision-driven embodied agents. arXiv preprint arXiv:2502.09560.
>
> [2] Tan, W., Zhang, W., Xu, X., Xia, H., Ding, Z., Li, B., ... & Lu, Z. (2024). Cradle: Empowering foundation agents towards general computer control. arXiv preprint arXiv:2403.03186.
>
> [3] Team, B. R., Cao, M., Tan, H., Ji, Y., Chen, X., Lin, M., ... & Zhang, S. (2025). Robobrain 2.0 technical report. arXiv preprint arXiv:2507.02029.

---

### Author Response · Authors · 2025-12-03
**Summary of Rebuttal**

Dear ACs/PCs/SACs,

We understand that due to the recent incident, we sincerely appreciate the time and effort dedicated by all reviewers and by the previous AC to this paper. To facilitate your quick review of the current status, we provide a summary of previous reviewers' comments on our work and our rebuttal updates, as well as the contribution of our paper.

First of all, we want to summarize the advantages of our work according to reviewers:

1. RoboMemory improves the long-term planning capability of embodied agents.

2. RoboMemory explores how to combine a comprehensive memory module with an embodied agent.

3. RoboMemory designs a comprehensive memory module based on the human brain. The parallelized architecture ensures the efficiency of the comprehensive memory module.

4. RoboMemory is tested/evaluated in multiple environments and outperforms baselines on various benchmarks.

5. RoboMemory enables the deployment of embodied agent systems with a comprehensive memory module to real-world environments and tests it on multiple tasks.

Secondly, the reviewers point out several concerns about our work and our rebuttal. We summarize our feedback:

> Reviewer ct4h's main concerns are:

1. The lack of memory mechanisms in baselines may make the comparison unfair.

2. Comparing reasoning models.

3. The efficiency concern.

Our answer is:

1. The baseline we use includes basic memory mechanisms; however, their memory mechanisms are likely insufficient for completing long-term tasks.

2. The baseline we use (including agent structure and prompts) is designed for a non-reasoning model, so we limit our comparison to standalone VLM-Agents and VLM-Agent Frameworks. The baseline result is sufficient to prove the advantage of our memory systems in long-term embodied tasks.

3. We clarify that we use multiple techniques, including parallelism, to improve efficiency.

4. We have added EB-Habitat to the main result table to address this concern. See Table 1. Meanwhile, we added several experiments to address the reviewer's concern. We introduce RoboOS [1] as a new baseline, featuring a powerful VLM design for Embodied tasks (RoboBRain2-32B), to address the concern about a lack of baseline.

> Reviewer Ni9E's main concerns are:

1. Lacks a baseline in real-world environments.

2. Experiments on tasks that require memory other than spatial memory are missing.

3. Need to compare with baseline, including spatial memory.

Our answer is:

1. We add the RoboOS [1] framework for both virtual environments and real-world environments as a baseline. The result is updated in our paper.

2. The benchmark in EB-Habitat and EB-ALFRED tests the comprehensive skill of the memory system beyond spatial memory. The ablation study validates this capability.

> Reviewer DhZZ's main concerns are:

1. Lack discussions about the hardware, costs and compute required for their approach against the baselines methods.

2. The real-world evaluation is done at a small scale and does not perform well.

3. No comparison with Baseline in the real environment.

Our answer is:

1. We acknowledge that while our parallelization design significantly boosts inference speed and system efficiency, it does not reduce the total computational cost (FLOPs). Our primary optimization goal is real-time performance and responsiveness rather than raw compute reduction. Furthermore, our framework is model-agnostic, allowing the flexibility to substitute smaller, more efficient models to balance computational costs as needed.

2. We acknowledge the limitations of the real-world experiments. However, testing long-horizon tasks containing both navigation and manipulation on mobile robots is challenging. Consequently, the sample size of real-world evaluations is inherently constrained. However, we conduct numerous experiments in a virtual benchmark (200 tasks) instead.

3. As a response to reviewer Ni9E, we add RoboOS as our baseline in real-world environments.


> Reviewer tMhd's central concerns:

1. The paper is written on a relatively high level, and the organization should be improved.

2. The baselines need to be strengthened by more recent and robotic-focused methods.

Our answer is:

1. We rewrite the 3.2 section in methodology and included pseudocode algorithms to provide greater detail to give more detail and improve the organization of our paper.

2. As our response to Ni9E, we add RoboOS as our baseline.

Ultimately, we would like to highlight the key contributions of our work. Firstly, we propose a brain-inspired, comprehensive memory system designed to enhance the long-term planning capability of an embodied agent.  Secondly, we introduce a memory system with multiple types of memory that is more efficient due to its highly parallelized design. Most importantly, we deploy an embodied agent with various memory modules in multiple virtual and real-world environments, demonstrating superior performance compared to baseline methods.

---

### Meta-Review · Area_Chair_sbFd · 2026-01-11

**Summary:**

The paper reports strong gains on EmbodiedBench and a small real-world robot study (ct4h; tMhd; DhZZ). Review discussions focused on the fairness and strength of virtual benchmark comparisons (ct4h; tMhd; Ni9E), the level of methodological detail and clarity of module interactions (tMhd), and the characterization of compute cost and real-world evidence (DhZZ; Ni9E). In rebuttal, the authors clarified baseline memory usage (ct4h), added RoboOS as a stronger baseline in both virtual and real-world settings (ct4h; Ni9E; DhZZ; tMhd), moved Related Work to the main paper (Ni9E), expanded Habitat results into the main table (ct4h), and rewrote the methodology with pseudocode while fixing a critic-related inconsistency (tMhd).

**Reviewer Concerns:**

Concerns largely addressed include clarification of baseline memory usage (including multi-frame Cradle) (ct4h), the addition of RoboOS as a stronger robotics-focused baseline in both virtual and real-world settings (ct4h; Ni9E; tMhd), relocation of Related Work to the main text (Ni9E), inclusion of Habitat results (ct4h), and improved methodological clarity via restructuring and pseudocode (tMhd).

Remaining issues may include limited comparison with explicit reasoning models (beyond RoboBrain2-32B) (ct4h), lack of uncertainty or multi-seed reporting despite deterministic evaluation (ct4h), doubts about EB-ALFRED’s demand for non-spatial memory (Ni9E), and high compute cost—parallelism improves latency but not FLOPs—along with modest real-world success rates (DhZZ).

**Reviewer Scores:**

ct4h: likely to increase from 4 to 5 after baseline and Habitat additions.

Ni9E: would not be changed much as skepticism about non-spatial memory may persist.

DhZZ: some possibility to increase the score after adding real-world baselines and clarifying cost, despite scale and success-rate concerns.

tMhd: some possibility to increase the score following added algorithmic detail, pseudocode, organizational improvements, and stronger baselines.

---

### Decision · Program_Chairs · 2026-01-26

Reject